# Orally delivered toxin–binding protein protects against diarrhoea in a murine cholera model

Marcus Petersson [1,2,10], Franz G. Zingl [3,4,5,10], Everardo Rodriguez-Rodriguez [2], Jakob K. H. Rendsvig [2], Heidi Heinsøe[2], Emma Wenzel Arendrup [2], Natalia Mojica[6], Dario Segura Peña [7], Nikolina Sekulić[7,8], Ute Krengel [6], Monica L. Fernández-Quintero[1], Timothy P. Jenkins [1], Lone Gram [1], Matthew K. Waldor [3,4,5,9], Andreas H. Laustsen [1,2] ✉ & Sandra Wingaard Thrane [2] ✉

The ongoing seventh cholera pandemic, which began in 1961, poses an escalating threat to public health. There is a need for new cholera control measures, particularly ones that can be produced at low cost, for the one billion people living in cholera-endemic regions. Orally delivered $V_H$Hs, functioning as target-binding proteins, have been proposed as a potential approach to control gastrointestinal pathogens. Here, we describe the development of an orally deliverable bivalent $V_H$H construct that binds to the B-pentamer of cholera toxin, showing that it inhibits toxin activity in a murine challenge model. Infant mice given the bivalent $V_H$H prior to *V. cholerae* infection exhibit a significant reduction in cholera toxin–associated intestinal fluid secretion and diarrhoea. In addition, the bivalent $V_H$H reduces *V. cholerae* colonization levels in the small intestine by a factor of 10. This cholera toxin–binding protein holds promise for protecting against severe diarrhoea associated with cholera.

The severe diarrhoeal disease cholera is caused by the comma-shaped Gram-negative bacterium *Vibrio cholerae*. The pathogen is readily transmitted through contaminated water or food[1]. After ingestion, bacteria proliferate in the small intestine (SI), where they initiate the production of cholera toxin (CTX)[2–4]. CTX is an AB$_5$ toxin comprised of five B-subunits (CTXB), which bind to the human ganglioside receptor GM1 with high affinity ($K_D = 0.73$ nM), and an enzymatically active A-subunit (CTXA)[5–7]. Upon release into the cytosol, CTXA stimulates cyclic adenosine monophosphate (cAMP) production which leads to chloride ion efflux across the epithelial cell membrane and ultimately to

fluid secretion observed as severe watery diarrhoea[6,8]. CTX also promotes release of nutrients into the lumen of the SI and thus enhances *V. cholerae* proliferation[9,10]. The sequence of CTXB has been conserved in *V. cholerae* evolution, and there are only two amino acid substitutions in the mature CTXB protein between the two O1 serogroup biotypes (classical and El Tor) that have caused pandemic cholera[4,11,12].

In 2022, the ongoing seventh cholera pandemic surged, with cholera spreading to new countries and an increased number of cholera cases[13]. In parallel, limitations in global resources to prevent cholera led to insufficient supplies of oral cholera vaccines (OCVs) and a

[1]Department of Biotechnology and Biomedicine, Technical University of Denmark, Kongens Lyngby, Denmark. [2]Bactolife A/S, Copenhagen, Denmark. [3]Department of Immunology and Infectious Diseases, Harvard T. H. Chan School of Public Health, Boston, MA, USA. [4]Division of Infectious Diseases, Brigham and Women's Hospital, Boston, MA, USA. [5]Department of Microbiology, Harvard Medical School, MA Boston, USA. [6]Department of Chemistry, University of Oslo, Oslo, Norway. [7]Centre for Molecular Medicine Norway, University of Oslo, Oslo, Norway. [8]Department of Molecular Medicine, Institute of Basic Medical Sciences, Faculty of Medicine, University of Oslo, Oslo, Norway. [9]Howard Hughes Medical Institute, MD Bethesda, USA. [10]These authors contributed equally: Marcus Petersson, Franz G. Zingl. ✉e-mail: ahola@bio.dtu.dk; swt@bactolife.com

change from the recommended two-dose OCV regimen to a one-dose regimen[13,14]. The annual deployment of OCVs (~23 million doses) constitutes only a fraction of what is needed to safeguard the one billion people at risk of cholera in developing countries[15,16]. OCVs have additional limitations, such as reduced efficacy in children less than five years of age and challenging supply-chain logistics (e.g., distribution and cold-chain requirements)[17,18].

New approaches are needed to ameliorate the increased global risk and burden of cholera. Orally administered single-domain antibodies (such as variable domain of heavy chain of heavy-chain only antibodies ($V_HH$s)) are promising candidates for neutralization of gastrointestinal (GI) pathogens[19]. These antibodies have several properties, including their high antigen specificity and stability in conditions found in the GI tract (i.e., low pH and a proteolytic environment) that make them particularly suited for oral applications in comparison with conventional antibody formats[20–23]. Moreover, the small size and simple structure of $V_HH$ constructs allows for highly efficient production in microbial cell factories, enabling large-scale, low-cost biomanufacture[24]. Earlier studies of anti-CTX milk immunoglobulins and oligosaccharides, as well as derivatives thereof, suggest that interfering with GM1 receptor binding abrogates the effect of CTX[25–28].

Here, we develop and characterize a bivalent $V_HH$ construct, BL3.2, that blocks the CTX–GM1 interaction by binding specifically to CTXB. We find that BL3.2 is stable in conditions relevant to the GI tract and attenuates CTX-induced cAMP production in human cells. Gavage of BL3.2 in infant mice infected with *V. cholerae* is effective in alleviating diarrhoea. Our findings suggest that orally delivered bivalent $V_HH$ constructs could be used as dietary supplements and help reduce the risk of cholera-induced diarrhoea. In turn, this could potentially help limit cholera outbreaks and transmission of *V. cholerae* in endemic regions.

## Results

### A bivalent $V_HH$ construct abrogates the CTXB–GM1 interaction

Two alpacas were initially immunized with CTXB, their serum collected and pooled, and their RNA isolated to construct a monovalent $V_HH$ library from which CTXB-specific $V_HH$s were identified using a phage display selection campaign. Of these $V_HH$s, 380 were expressed in *Escherichia coli*, enabling screening of supernatants for the ability to

abrogate the CTXB–GM1 interaction in a fluorescence-based immunoassay (Supplementary Fig. 1). Unique monovalent $V_HH$s that blocked CTXB–GM1 binding more than 25% were validated in a second fluorescence-based immunoassay at a defined $V_HH$:CTXB molar ratio of 10:1. The blocking capacity of the monovalent $V_HH$ BL3.1 was 88% against the CTXB–GM1 interaction, which was better than the other $V_HH$s tested (44–67% blocking capacity) (Fig. 1a). The CTXB-binding affinity of BL3.1 was determined through bio-layer interferometry (BLI) and surface plasmon resonance (SPR), using BL3.1 both as ligand and analyte in the two respective experiments (Supplementary Table 1 and Supplementary Fig. 2). BL3.1 displayed high affinity for CTXB, with an apparent $K_D$ determined to be 0.76 nM by BLI (with BL3.1 as ligand) and the monovalent $K_D$ to be 85 nM by SPR (with CTXB as ligand) (Supplementary Table 1). These values are similar to the $K_D$ (77 nM) of a previously reported anti-CTX $V_HH$[29].

The bivalent $V_HH$ construct BL3.2 was generated through genetic fusion ($V_HH-(G_4S)_3-V_HH$) of two monovalent BL3.1 subunits, a protein engineering approach previously shown to enhance target binding[30,31]. BL3.2 displayed complete blocking (100%) of the interaction between CTXB and GM1 at the lowest molar ratio possible (1:1 $V_HH-V_HH$:CTXB), which is a higher level of blocking than what was observed for the monomeric component alone (79%) at the same ratio (Fig. 1b). The dimeric BL3.2 was therefore used in the studies described below.

### BL3.2 is stable under GI passage–relevant conditions

Biochemical conditions representative of passage through the stomach and SI, the site of *V. cholerae* colonization and CTX secretion, were generated through incubation of BL3.2 at 37 °C in simulated gastric fluid (SGF; pH 1.2) and simulated intestinal fluid (SIF; pH 6.8). CTXB-binding activity of BL3.2 was investigated following incubation in either phosphate-buffered saline (PBS), SGF, or SIF for a maximum of 4 h (SGF) to 5 h (SIF), representative of the times documented for in vivo gut transit[32]. BL3.2 displayed no reduction in CTXB-binding activity after 4 h incubation in SGF or 5 h in SIF, relative to a BL3.2 control kept at 4 °C in PBS (Fig. 2a). Protein thermal stability screening using differential scanning fluorimetry (Protein Thermal Shift™) showed that BL3.2 had comparable thermal stability (72 °C) to CTX (68 °C) (Fig. 2b and Supplementary Fig. 3).

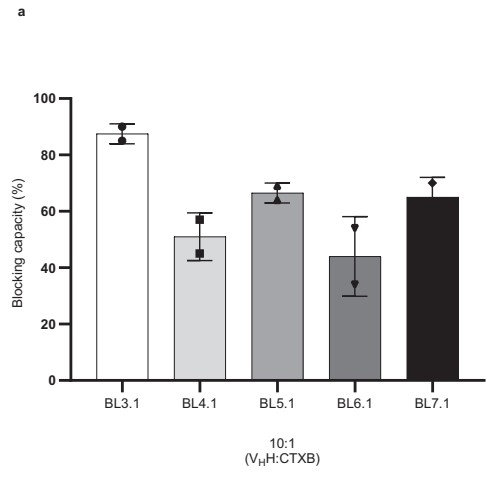

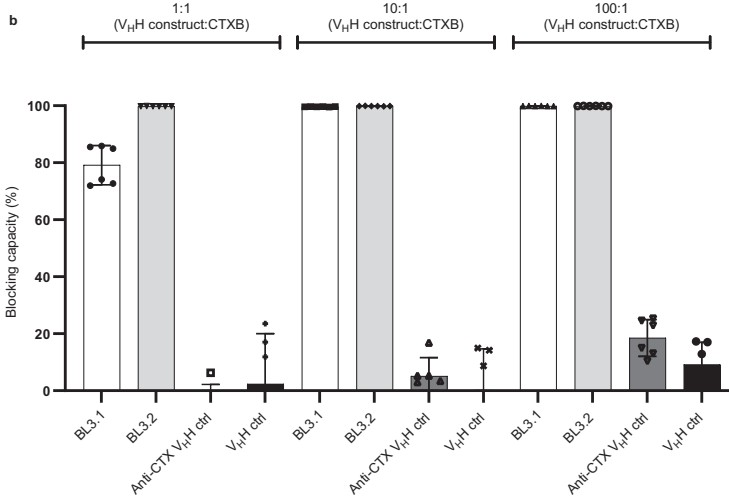

**Fig. 1 | Blocking capacity of the $V_HH$ constructs against the CTXB–GM1 interaction. a** The ability to block the CTXB–GM1 interaction for selected monomeric $V_HH$s at a molar ratio of 10:1 ($V_HH$:CTXB). The average blocking effect of each $V_HH$ was calculated from a single measurement of technical duplicates normalized against a control mixture containing CTXB and a $V_HH$ without specificity for CTXB. Error bars represent standard deviation. Source data are provided as a Source Data file. **b** The CTXB-blocking capacity of the monovalent BL3.1 in comparison to the bivalent ($V_HH-(G_4S)_3-V_HH$) BL3.2 at molar ratios of 1:1, 10:1, and 100:1 ($V_HH$ construct:CTXB). A previously reported anti-CTX $V_HH$ control and a negative $V_HH$ control without specificity for CTXB were included as well[42]. The average blocking capacity was calculated from duplicate measurements of technical triplicates. Error bars represent standard deviation. Source data are provided in a Source Data file.

a

b

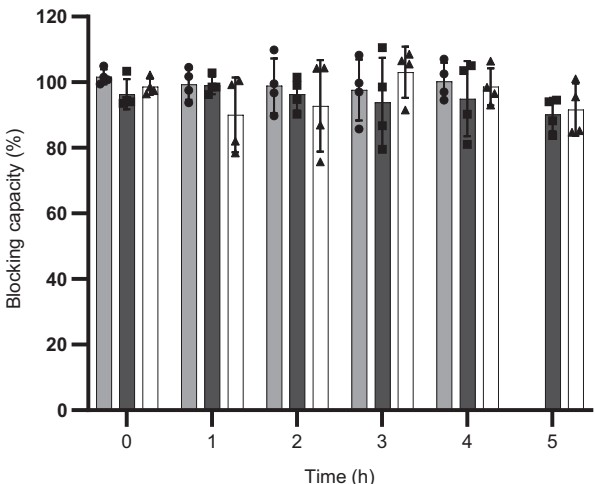

**Fig. 2 | Thermal and simulated gastrointestinal stability profile of BL3.2.**
**a** CTXB-binding activity of BL3.2 after incubation for up to 4 h in simulated gastric fluid (SGF; pH 1.2) or up to 5 h in simulated intestinal fluid (SIF; pH 6.8) at 37 °C. Binding activity was normalized against a BL3.2 control stored at 4 °C in phosphate-buffered saline (PBS) for 5 h. The average CTXB-binding activity is based on measurements from two experiments of one sample per condition (SGF, SIF, and PBS) analysed in technical duplicates. Error bars represent standard deviation. Source data are provided in a Source Data file. **b** Thermal denaturation temperature ($T_m$) of BL3.2 and CTX in PBS. Each sample was analysed in triplicates, with lines indicating the averages.

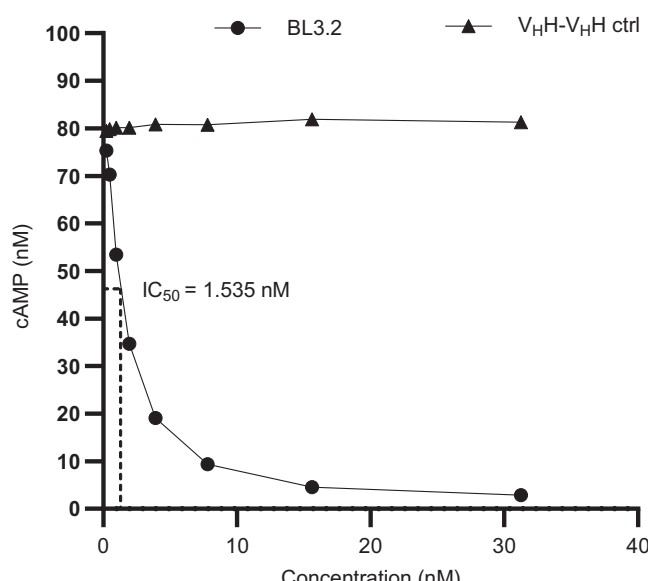

**Fig. 3 | Functional neutralization of CTX by BL3.2 in a human colon adeno-carcinoma cell assay.** Reduction of intracellular cyclic adenosine monophosphate (cAMP) production in HCA-7 cells incubated with CTX (0.115 nM) pre-mixed with increasing concentrations (0.240–31.25 nM) of the bivalent $V_H$H construct BL3.2 (circles) or a non-specific bivalent $V_H$H construct control (triangles). Levels of intracellular cAMP were interpolated from a sigmoidal four parameter logistic cAMP standard curve ($R^2 = 0.9543$) based on triplicate measurements. Each data point represents an interpolated mean value from biological duplicates comprised of three technical replicates. Source data are provided in a Source Data file.

## BL3.2 inhibits CTX activity in a human cell–based assay

The ability of BL3.2 to block the key cellular consequence of CTX–GM1 interaction was investigated using bioluminescent detection of intracellular cAMP in human colon adenocarcinoma (HCA-7) cells incubated with a mixture of BL3.2 and CTX. Different concentrations (0.240–31.25 nM) of BL3.2 were pre-mixed with a fixed concentration (0.115 nM) of CTX before this mix was added to HCA-7 cell monolayers. In contrast to a previously reported CTX-specific $V_H$H, BL3.2 achieved complete CTX neutralization in a concentration-dependent manner (Supplementary Fig. 4). BL3.2 inhibition of intracellular cAMP production by toxin neutralization was equivalent to a > 27-fold decrease of cAMP in comparison to a bivalent $V_H$H control that does not bind CTX (Fig. 3). BL3.2 achieved half-maximal relative inhibitory concentration ($IC_{50}$) at 1.535 nM. This $IC_{50}$ corresponds to a $V_H$H–$V_H$H:CTX molar ratio of 13:1, or approximately 5:1 when considering the number of CTXB binding sites for each bivalent BL3.2 molecule ($V_H$H–$V_H$H:CTXB).

## BL3.2 binds to the conserved GM1-binding pocket of CTXB

Predictions based on ColabFold in combination with classical molecular dynamics identified four potential BL3.2–CTXB protein–protein interfaces (Model 1–4) (Supplementary Table 2). Based on clustering and interaction energies (electrostatic and van der Waals) together with available structural data on CTX–GM1 complexes, Model 4 was considered the most stable conformational arrangement (Fig. 4a). Two amino acid residues (Asp30 and Asp31) in the BL3.2 complementarity-determining region one (CDR1), two (Asp55 and Ser57) in CDR2 and five (Tyr102, Asn104, Ser105, Gln107, and Asp111) in CDR3 were critical for CTXB binding and interfer with CTX–GM1 interaction based on molecular dynamics simulations of Model 4 (Fig. 4b). Similarly, nine primary amino acids in the CTXB epitope (His13, Asn14, Ser55, Gln56, His57, Asp59, Gln61, Trp88, and Lys91) and five amino acids in the adjacent CTXB in the pentamer (Lys34, Arg35, Glu36, Ser55, and Gln56) were predicted to play a crucial role in interacting with BL3.2 (Fig. 4c). The interacting residues identified via ColabFold are in good agreement (root mean square deviation (RMSD) the Cα of the two complexes <1.5 Å) with the predicted BL3.2–CTXB interaction site identified by the independent EpiC machine learning platform (Supplementary Fig. 5).

The predicted BL3.2–CTXB interactions were experimentally validated by hydrogen–deuterium exchange mass spectrometry (HDX–MS) and size-exclusion chromatography (SEC) analysis. The greater part of the CTXB epitope predictions made by the two

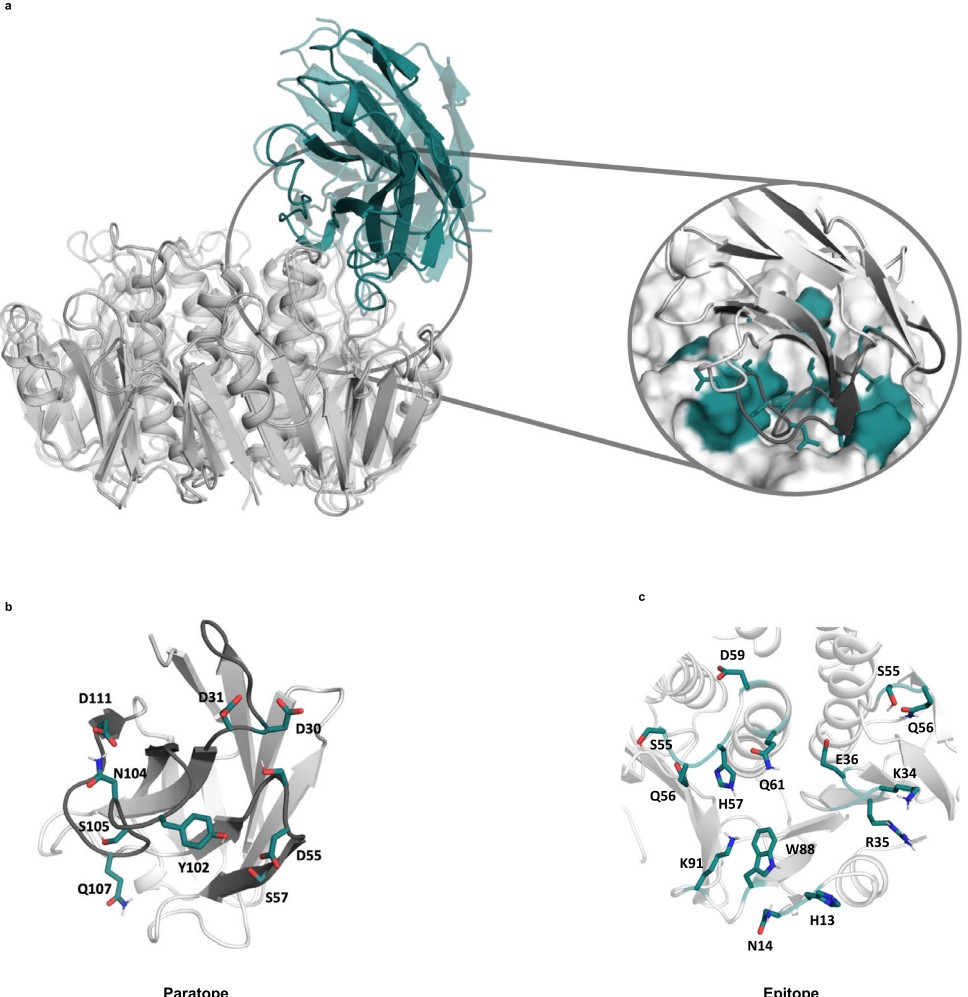

**Fig. 4 | Predicted interface between BL3.2 and CTX. a** The most stable conformational binding arrangement between BL3.2 (green) and CTXB pentamer (grey) based on ColabFold and molecular dynamics simulations. **b** The nine amino acid residues in the complementarity-determining region one (CDR1) (Asp30 and Asp31), CDR2 (Asp55 and Ser57), and CDR3 (Tyr102, Asn104, Ser105, Gln107, and

Asp111) of the BL3.2 paratope predicted to be essential for CTX binding. **c** The 14 amino acids in the CTXB epitope identified in silico to be crucial for BL3.2 interaction, nine in the primary CTXB (His13, Asn14, Ser55, Gln56, His57, Asp59, Gln61, Trp88, and Lys91) and five from the adjacent subunit in the pentamer (Lys34, Arg35, Glu36, Ser55, and Gln56).

independent machine learning platforms are concordant with HDX–MS analysis of BL3.1–CTXB interaction (Fig. 5). Two distinct regions of CTXB exhibited HDX protection upon binding to BL3.1; the first region comprised amino acid positions 29–38, and the second region comprised positions 50–66 (Fig. 5a and Supplementary Fig. 6, 7). Mapping of the HDX-MS results onto the CTXB holotoxin structure indicates that the CTXB epitope recognized by BL3.1 is a conformational epitope, consisting of two spatially adjacent loops of the CTXB pentamer involved in binding the terminal galactose in GM1 (Fig. 5b and c). Both of these loops contribute with key amino acid residues (in positions 33, 55–57, and 61) in the CTX–GM1 interaction (Supplementary Fig. 8)[5,33]. These amino acid residues are conserved among the three *ctxB* genotypes (*ctxB1*, *ctxB3*, and *ctxB7*) known to have caused cholera pandemics to date[34]. Positions 11–14, 88, and 90-91 of CTXB were not identified as a part of the BL3.2 epitope by HDX–MS. Since region 11–14 is covered with three different peptides that have the same rate of H/D exchange in the presence and absence of BL3.1, it is likely that this region is not recognized by the $V_HH$ construct (Supplementary Fig. 9). However, no high-quality peptides were obtained for residues 88, 90, and 91, which prevents a confident assessment of their recognition by BL3.1. Furthermore, binding of BL3.2 to CTXB resulted in the formation of multiple oligomeric species (Supplementary Fig. 10). This may be attributable to multiple BL3.2

molecules binding to a single CTXB pentamer or through BL3.2 engaging separate CTXB pentamers at its two binding sites.

In addition to experimental verification of the toxin epitope, amino acid mutations of the predicted BL3.2 paratope were made to demonstrate their effect on CTXB binding. A single amino acid mutation (Y102I) in the predicted CDR3 region of BL3.2 resulted in more than 95% reduction of BL3.1 blocking of the CTXB–GM1 interaction at a 50:1 molar ratio ($V_HH$:CTXB) (Supplementary Fig. 11). An additional mutation in the same region (Y102I and N104L), completely abolished the CTX-binding capacity of BL3.1 in the molar ratio range of 50:1 to 1000:1 ($V_HH$:CTXB) (Supplementary Fig. 11).

## Orally delivered BL3.2 inhibits CTX activity in vivo

The ability of BL3.2 to block CTX activity in vivo was assessed using 5-day-old CD-1 mice, a widely used model of cholera and the recommended model for targeted (e.g., CTX) investigations[35,36]. In a first experiment, BL3.2 (9 mg ml⁻¹) was pre-incubated at 37 °C for 30 min with CTX (1 mg ml⁻¹) before a single (50 μl) oral gavage of the mixture was administered. In a control group ($n = 14$), animals were inoculated with a mixture of bovine serum albumin (BSA; 9 mg ml⁻¹) and CTX (1 mg ml⁻¹). The BL3.2 group had significantly less weight loss than control animals ($P < 0.05$ with the two-tailed Mann–Whitney U test),

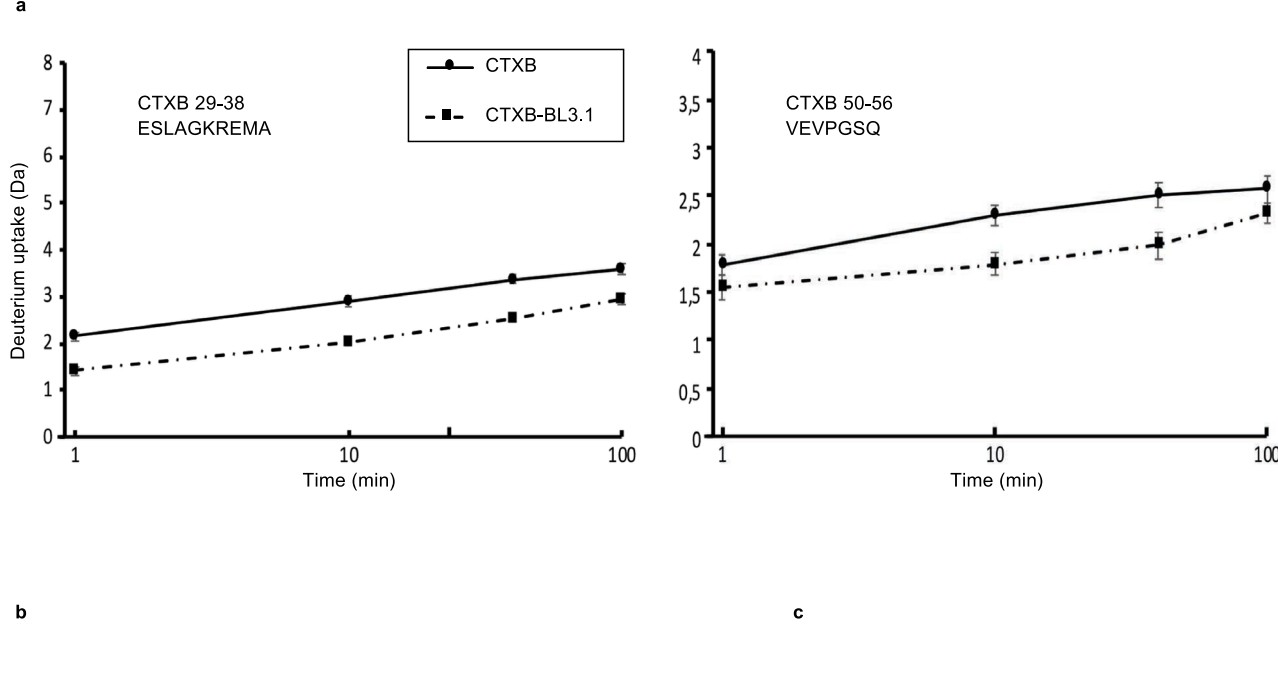

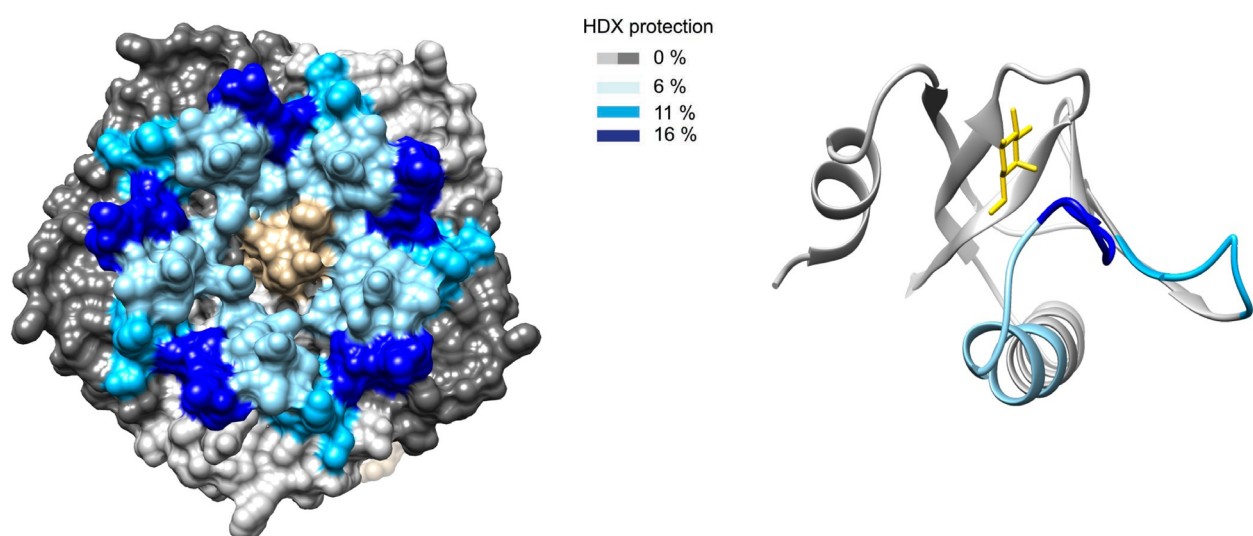

**Fig. 5 | Hydrogen–deuterium exchange mass spectrometry (HDX–MS) analysis of the CTX epitope of BL3.1. a** The level of HDX for CTXB alone and CTXB bound to BL3.1 with regards to two specific regions of the toxin: positions 29–38 and positions 50–56. The solid and dashed lines show the deuterium incorporation for CTXB and the CTX–BL3.1 complex, respectively. Each time point (1, 10, 40 and 100 min) was analysed in technical triplicates. Data are presented as mean values with standard deviation. **b** Surface representation of the CTXB pentamer. Shades of grey represent no HDX differences upon binding BL3.1. The level of HDX protection upon binding BL3.1 is indicated with different shades of blue. **c** A ribbon representation of a single subunit of CTXB with mapped HDX protection in shades of blue. A galactose molecule is shown in yellow, indicating the site of interaction with the intestinal cell receptor GM1.

suggesting that BL3.2 binding to CTX in vitro is sufficiently robust to neutralize the effects of the toxin in vivo (Supplementary Fig. 12).

To evaluate the potential protective effect of BL3.2, the in vivo CTX-neutralizing capacity of BL3.2 was directly evaluated when the antibody construct was administered separately from CTX. Two oral administrations (9 mg ml$^{-1}$ each) of BL3.2 (or BSA) were given, one administration 1 h prior to and one administration 3 h after CTX (0.1 mg ml$^{-1}$) administration to ensure BL3.2 presence in the GI tract (Fig. 6a). Severe diarrhoea was observed 9 h post-delivery of CTX among all mice ($n = 5$) in the control group (Fig. 6b). In contrast, none of the mice that received BL3.2 ($n = 5$) exhibited diarrhoea. In addition, mice administered BL3.2 showed significantly less weight loss and CTX-associated intestinal fluid

secretion in the SI in comparison to the BSA control group (Fig. 6b, c, $P < 0.05$ and $P < 0.01$, respectively, with the two-tailed Mann–Whitney U test).

## BL3.2 reduces intestinal colonization levels of *V. cholerae*

To evaluate the neutralizing capacity of BL3.2 against CTX produced in vivo during infection by *V. cholerae*, 5-day-old CD-1 mice were orally administered either BL3.2 (9 mg ml$^{-1}$) or BSA (9 mg ml$^{-1}$) twice, once 1 h before orogastric challenge with $2.8 \times 10^8$ colony-forming units (CFU) of a 2022 virulent *V. cholerae* clinical isolate and once 5 h after (Fig. 7a). Animals given BL3.2 ($n = 6$) had significantly ($P < 0.01$ with the two-tailed Mann-Whitney U test) reduced weight loss (Fig. 7b) and

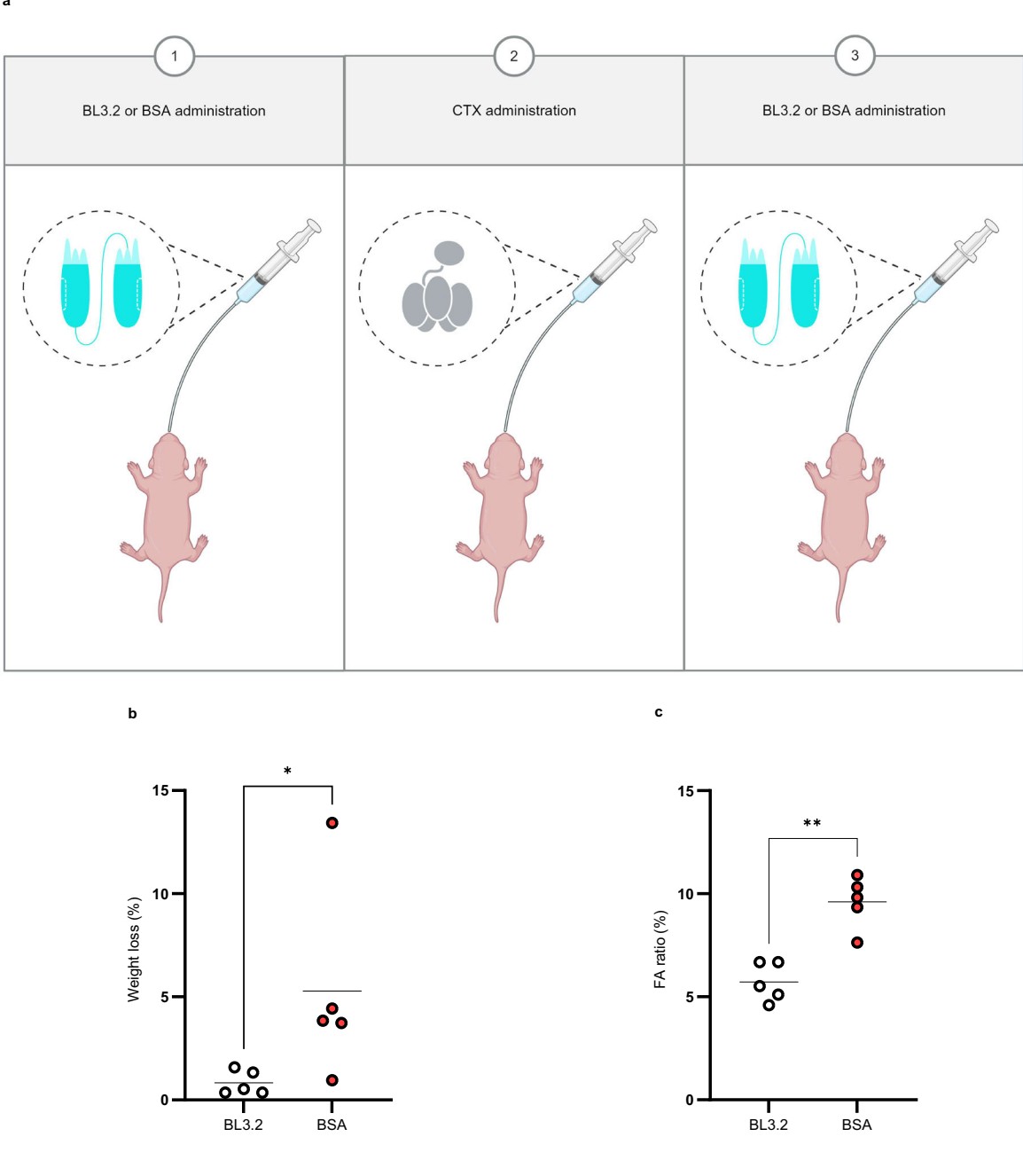

**Fig. 6 | CTX-neutralizing capacity of BL3.2 in infant mice. a** Schematic representation of the two oral administrations of BL3.2 given to 5-day-old CD-1 mice (Charles River Laboratories, strain 022, mixed sex); 1 h before and 3 h after CTX administration. Created using BioRender. Laboratory, T. (2025) https://BioRender.com/c95v770. **b** Impact of the orally delivered $V_H$H construct BL3.2 on the severity of CTX-associated diarrhoea (weight loss) in infant mice. Mice were given two oral administrations of either BL3.2 ($n = 5$, 9 mg ml$^{-1}$) or bovine serum albumin (BSA) as a control ($n = 5$, 9 mg ml$^{-1}$); one 3 h prior to oral delivery of CTX and one 3 h after oral delivery of CTX. diarrhoeal onset (red) or diarrhoeal absence (white) was visually monitored up until the experiment was terminated, 9 h following CTX administration. Horizontal lines indicate median weight loss for BL3.2 (0.53%) and BSA (3.8%), and statistical significance (*$p < 0.05$) was calculated using the two-tailed Mann–Whitney U test ($P = 0.0317$). Source data are provided in a Source Data file. **c** Impact of the orally delivered bivalent $V_H$H construct BL3.2 on CTX-induced fluid accumulation (FA) in the small intestine of infant mice. Mice were given two oral administrations of either BL3.2 ($n = 5$, 9 mg ml$^{-1}$) or BSA as a control ($n = 5$, 9 mg ml$^{-1}$), as previously described. Statistical difference (**$P < 0.01$) between median FA ratio (horizontal line) for BL3.2 (5.5%) and BSA (9.8%) was calculated using the two-tailed Mann–Whitney U test ($P = 0.0079$). Source data are provided in a Source Data file.

intestinal fluid accumulation (Fig. 7c) compared to animals in the BSA control group ($n = 6$). Furthermore, there were approximately 10-fold fewer *V. cholerae* CFU recovered from intestinal homogenates of mice given BL3.2 ($1.6 \times 10^8$ CFU) compared with the BSA control group ($1.8 \times 10^9$ CFU) 22 h after infection (Fig. 7d).

## Discussion

Recent efforts to create alternative cholera control measures have focused mainly on the development of new vaccine candidates, such as the oral cold-chain-free vaccine MucoRice, the probiotic-like vaccine PanChol, or monoclonal antibodies against CTX (7A12B3

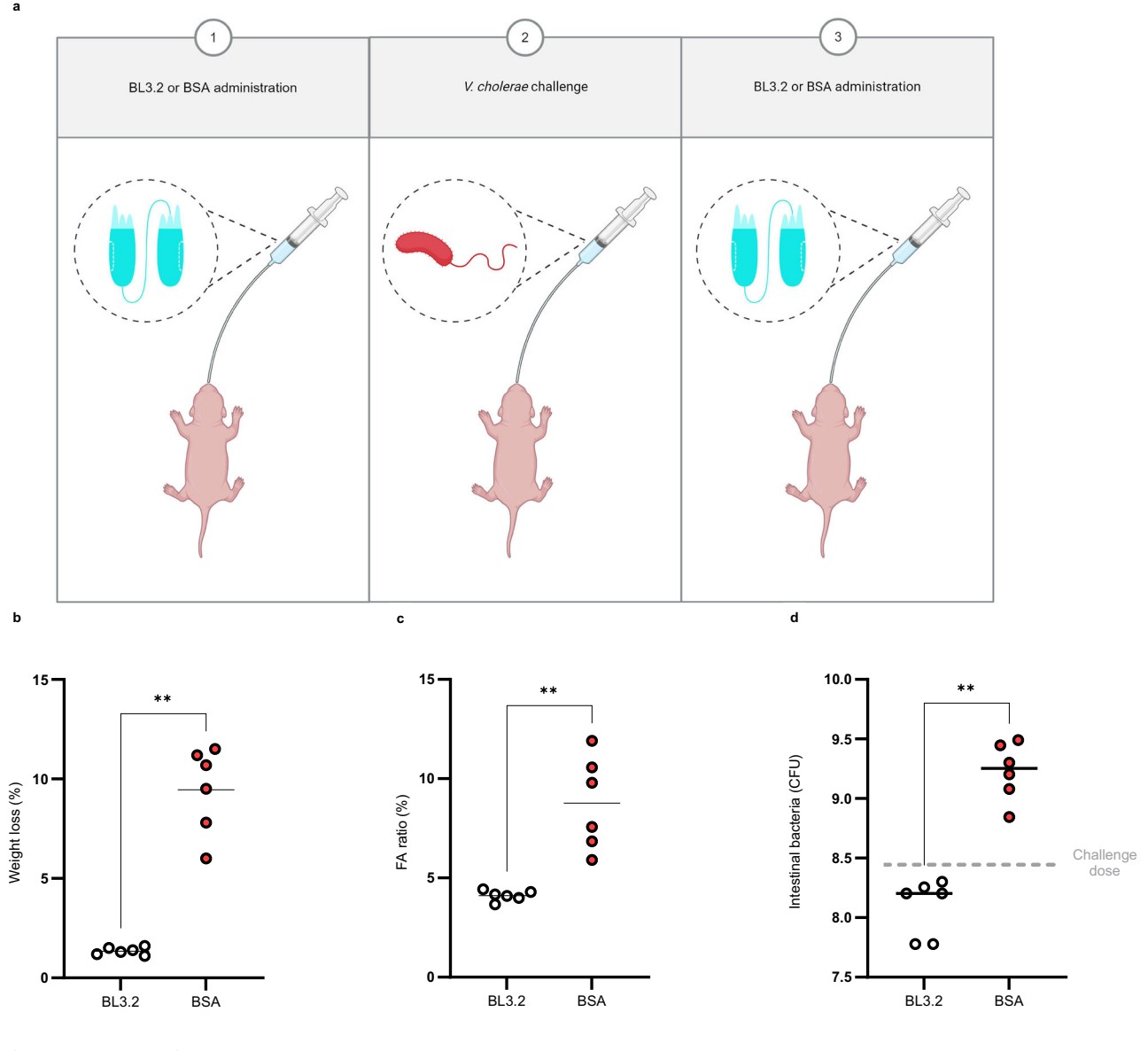

O  No diarrhea    ● Diarrhea

**Fig. 7 | In vivo evaluation of BL3.2 against clinical isolate of *V. cholerae*.**
**a** Schematic representation of the two oral administrations of BL3.2 given to 5-day-old CD−1 mice (Charles River Laboratories, strain 022, mixed sex); 1 h before and 5 h after orogastric challenge with a *V. cholerae* clinical isolate. Created using BioRender. Laboratory, T. (2025) https://BioRender.com/b75t379. **b** Weight loss of infant mice 22 h after orogastric inoculation with *V. cholerae* ($2.8 \times 10^8$ CFU), given two oral administrations of either BL3.2 ($n = 6$, 9 mg ml$^{-1}$) or bovine serum albumin (BSA) ($n = 6$, 9 mg ml$^{-1}$); 1 h before challenge with *V. cholerae* and 5 h after. Median for diarrhoea-induced weight loss (diarrhoea in red and no diarrhoea in white) determined to be 1.4% for BL3.2 and 10% for the BSA control. Statistical difference (**$P < 0.01$) estimated using the two-sided Mann−Whitney U test ($P = 0.0022$). Source data are provided in a Source Data file. **c** Fluid accumulation (FA) ratio (% of body weight) determined for infant mice challenged with *V. cholerae* ($2.8 \times 10^8$

CFU), given two oral administrations of either BL3.2 ($n = 6$, 9 mg ml$^{-1}$) or BSA ($n = 6$, 9 mg ml$^{-1}$); one hour before challenge with *V. cholerae* and five hours after. Median for FA ratio (diarrhoea in red and no diarrhoea in white) determined to 4.1% for BL3.2 and 8.7% for the BSA control. Statistical difference (**$P < 0.01$) estimated using the two-sided Mann−Whitney test ($P = 0.0022$). Source data are provided in a Source Data file. **d** The number of *V. cholerae* found in the small intestine of infant mice 22 h after orogastric inoculation with $2.8 \times 10^8$ CFU (dashed line), given two oral administrations of either BL3.2 ($n = 6$, 9 mg ml$^{-1}$) or BSA ($n = 6$, 9 mg ml$^{-1}$); 1 h before challenge with *V. cholerae* and 5 h after. Average CFU per organ (solid line) determined to be $1.6 \times 10^8$ CFU for BL3.2 and $1.8 \times 10^9$ CFU for the BSA control. Statistical difference (**$P < 0.01$) estimated using the two-sided Mann−Whitney U test ($P = 0.0022$). Source data are provided as a Source Data file.

and 9F9C7)[37–39]. However, it has become evident that future anti-cholera strategies will require more proactive, rapid, and efficiently distributed countermeasures[40]. Here, we describe the development of a bivalent V$_H$H construct, BL3.2, capable of abrogating the CTX−GM1 receptor interaction and reducing *V. cholerae* enterotoxicity in the GI tract when the protein was administered orally to infant mice. In infant mice given BL3.2 prior to challenge with *V. cholerae*, there were significant reductions in CTX-associated

weight loss, intestinal fluid secretion, and pathogen burden in the SI.

We used a GM1 receptor-blocking dissociation-enhanced lanthanide fluorescence immunoassay (DELFIA) to screen a panel of monovalent V$_H$Hs in *E. coli* culture supernatants, which allowed for high-throughput functional screening and identification of several V$_H$H hits with CTX−GM1-blocking capacity. Only a few antibodies able to abrogate the CTX−GM1 interaction have been reported to date, potentially

due to the lack of early functional screening in the selection process[39,41,42]. In previous studies, traditional immunosorbent assays (i.e., CTX- or GM1-capture) for primary selection were employed, before more advanced (e.g., competitive ELISA) assays were utilized in the later stages of antibody characterization[39,42]. In contrast, early in vitro screening using receptor-blocking assays has been successfully employed for the rapid identification of toxin-neutralizing antibody constructs in other research areas[43,44]. This highlights the importance of early functional screening, together with optimized single-domain antibody selection campaigns, for the future development of orally delivered toxin-neutralizing $V_H H$ constructs[45]. We also argue that early (in vitro) assessment of the developability profile for such constructs can help identify leads that are fit-for-purpose (i.e., functional under GI conditions and suitable for oral delivery)[46,47].

We found that the oral delivery of BL3.2 reduced the burden of *V. cholerae* in the SI along with limiting fluid secretion in vivo. In a previous study, a *V. cholerae* mutant unable to produce CTX was utilized to demonstrate that the toxin is necessary for the pathogen's nutrient (i.e., heme and long-chain fatty acids) acquisition and growth[10]. We hypothesize that diminished *V. cholerae* proliferation in vivo explains the reduced pathogen burden; BL3.2's blockade of CTX-mediated release of nutrients into the SI lumen could impair the pathogen's normal replication in vivo[9,10,48]. While animal studies provide supportive evidence, the cohorts are relatively small (5 or 6 animals), and more studies are warranted. To better elucidate the mechanism by which BL3.2 limits *V. cholerae* colonization, a $V_H H$ construct without specificity for the GM1 receptor-binding site of CTXB should be included.

Other approaches, including plant extracts and nanoparticles, have been investigated for their capacity to neutralize *V. cholerae* toxicity in the GI tract[49–52]. Several different plant extracts (e.g., raspberry leaves and grape seeds) have been shown to reduce intracellular CTX uptake in vitro, and their wide use in traditional medicine could indicate a favourable safety profile for human applications[49,50]. Yet, the target-specificity for anti-toxin compounds found in plants, as well as their exact mechanism of action, remains to be defined[49,50]. Similarly, orally delivered nanoparticles have been shown to diminish CTX-induced HCA-7 cell production of cAMP 3-fold and to significantly reduce *V. cholerae* colonization levels[51,52]. However, the oral use of nanoparticles requires further studies to ensure safety due to their high bioavailability and potential long-term bioaccumulation given their abiotic nature, which could pose a toxicity risk[51–53]. In contrast, single-domain antibody constructs share similarities with natural immunoglobulins, such as IgA antibodies against CTX found in breast milk, and could therefore face fewer safety challenges[48,54].

The interface between BL3.2 and CTX predicted by machine learning modelling, and further substantiated by HDX−MS studies of the BL3.1−CTX complex, indicates that BL3.2 targets a CTXB epitope that has remained highly conserved throughout *V. cholerae* evolution. In particular, the GM1 receptor-binding site of CTXB continues to remain intact among recent clinical isolates, suggesting that BL3.2 targets an epitope that is unlikely to escape BL3.2 binding and that BL3.2 may therefore be suitable for long-term usage in cholera control programs[55,56]. It could also be speculated that the ability of $V_H H$s to control *V. cholerae* pathogenesis through neutralization of an extracellular soluble protein will enforce less evolutionary pressure on the development of *V. cholerae* resistance mechanisms compared to bactericidal countermeasures (e.g., antibiotics or bacteriophages)[57–59]. Regardless, our findings suggest that future household-based studies of healthy at-risk individuals are warranted to test if the results translate to humans.

$V_H H$ constructs have been demonstrated to possess notable stability under GI conditions both in vitro and in vivo[20,47,60]. As an example of their use for GI applications, three daily administrations of a similar $V_H H$ construct (35 mg kg$^{-1}$ per day) were shown to reduce rotavirus-associated diarrhoeal severity in infants, and co-administration with oral rehydration salts could enable a further lowering of the dose[61,62].

Similarly, piglets challenged with enterotoxigenic *E. coli* (ETEC) were fed 24.2 mg per day of a bivalent $V_H H$ construct against heat-labile enterotoxin (LT)[60]. The infant mouse model utilized in this study limits the total amount and number of daily administrations of BL3.2, which makes it difficult to extrapolate the human equivalent dose. Future studies should therefore aim to evaluate production cost of BL3.2 in relation to human dosing and formulation requirements. If reductions in diarrhoea and pathogen shedding are observed in humans, we anticipate that BL3.2-like proteins will not just benefit individuals at immediate risk of *V. cholerae* infection, but also reduce cholera transmission and thereby protect communities[9]. Ultimately, we envision that orally delivered anti-CTX $V_H H$ constructs will find utility in existing or novel fortified food products and offer a protective strategy for sustainable cholera control[61–63]. Given the high structural similarity between CTX and other $AB_5$-type toxins, it is plausible that similar approaches to the one presented here could find utility for other bacterial pathogens, such as LT-producing ETEC and perhaps even more distantly related toxin-producing pathogens[55,64].

## Methods

### Experimental animals and ethical considerations

Alpaca immunization and blood collection was performed by the VIB Nanobody Service Facility (Brussels, Belgium). Immunizations and handling of the animals were performed according to directive 2010/63/EU of the European parliament for the protection of animals used for scientific purposes and approved by the Ethical Committee for Animal Experiments of the Lamasté (permit No. 2020.1_NSF). All experiments involving mice were conducted in a biosafety level 2 (BSL2) facility at the Brigham and Women's Hospital (Boston, the Unites States). The animal experiments were performed according to a protocol (2016N000416) reviewed and approved by the Brigham and Women's Hospital Institutional Animal Care and Use Committee and in compliance with the Guide for the Care and Use of Laboratory Animals. Mice were housed under specific pathogen free conditions in a temperature (68–75 °F) and humidity-controlled (50%) facility with 12 h light/dark cycles, with unlimited food and water availability. Infant (5-day-old) CD-1 mice (Charles River Laboratories, strain 022) of mixed sex included in experimental and control groups were co-housed until 30 min before the experimental procedures began. Within 24 h after the start of the experiment, mice were euthanized by isoflurane inhalation followed by decapitation.

### Toxin production and biotinylation

For in vitro and in vivo evaluation of $V_H H$ constructs, including stability and affinity studies, commercially available CTX (Sigma-Aldrich, C8052) and CTXB (Sigma-Aldrich, C9903) were used. For the molar ratios presented in this study, CTXB denotes a single monomeric B-subunit (12 kDa), while CTXB pentamer refers to the pentameric assembly of B-subunits (60 kDa). CTX indicates the complete toxin (86 kDa), which includes the catalytic CTXA. For structural analysis of BL3.1/BL3.2 and toxin interaction (i.e., HDX−MS analysis and SEC), *ctxB* was cloned into a **pET21b(+)** plasmid and expressed in *E. coli* BL21 (DE3) for 14–18 h using 0.5 mM of isopropyl-β-d-thiogalactopyranoside (IPTG). Cells were harvested by centrifugation (6,900 g for 20 min) and the toxin was isolated from the periplasmic space by osmotic shock via treatment with sucrose buffer (Tris-HCl supplemented with 25% sucrose and 5 mM EDTA at pH 8.0) and periplasmic lysis buffer (5 mM $MgCl_2$ buffer with 150 µg ml$^{-1}$ lysozyme). CTXB was purified using a D-galactose-sepharose affinity column (Thermo Scientific) followed by SEC (Superdex 200 Increase 10/30 GL column) on an ÄKTA system (GE Healthcare). The toxin was stored in PBS (pH 7.4).

For toxin biotinylation, lyophilized CTXB was dissolved in PBS to a final concentration of 1 mg ml$^{-1}$. No-Weigh NHS-PEG$_4$-Biotin (Thermo Scientific, A39259) reconstituted in deionized water was added to CTXB at a toxin:biotinylation reagent ratio of 1:2 and incubated at room

temperature for 30 min. Several washing rounds with PBS were carried out using 3 kDa molecular weight cut-off (MWCO) protein concentrators (Thermo Scientific, 16311964) to reduce excess reagent at least 125-fold.

## Monovalent V$_H$H library generation

Two alpacas (Proton and Lloyd) were serially immunized at the VIB Nanobody Core (Brussels, Belgium) for a total of six times over seven weeks, with 80 µg of CTXB per immunization. At each timepoint, 100 ml of serum was collected, peripheral blood lymphocytes isolated, and total RNA of pooled sera was reverse transcribed into cDNA, which was cloned into the **pMECS** phagemid vector using *PstI* and *NotI* restriction enzymes. Over three rounds of panning against coated CTXB (100 µg ml$^{-1}$ in 100 mM NaHCO$_3$), CTXB-specific transformants were enriched from the generated libraries. After the first panning round, the phage outputs were pooled and the polyclonal phagemid vectors encoding CTXB-specific V$_H$H domains were used to infect *E. coli* TG1, plated on lysogeny broth (LB) agar plates supplemented with 50 µg ml$^{-1}$ of kanamycin, and used for another round of panning. A total of 380 randomly selected clones from the two final rounds of panning were subcloned into the **pSANG10** vector optimized for *E. coli* BL21(DE3) expression[20,65].

## V$_H$H constructs blocking capacity of CTXB–GM1 interaction

A total of 380 monovalent V$_H$H constructs were evaluated based on their ability to abrogate the interaction between the monosialoganglioside GM1 receptor and CTXB, using GM1 immobilized onto microplates[66]. Black 96-well Immuno Plates (Thermo Scientific, 437111) were coated (overnight) with 100 µl GM1 (Sigma-Aldrich, G7641) in PBS at a concentration of 5 µg ml$^{-1}$. Plates were washed three times with PBS and blocked with PBS supplemented with 3% non-fat milk (M-PBS). The 380 monovalent V$_H$Hs were expressed in 96-well plates, and 30 µl of each *E. coli* supernatant was incubated with biotinylated CTXB (36 nM) in M-PBS at 37 °C for 30 min before added to the GM1-plates and incubated at room temperature (1 hour). As a negative control, CTXB (36 nM) was incubated in the absence of any V$_H$H. After three washes with PBS-Tween (0.2%) and three washes with PBS, 100 ng ml$^{-1}$ of streptavidin-conjugated europium (PerkinElmer, 1244-360) diluted in DELFIA assay buffer (PerkinElmer, 4002-0010) was added, and DELFIA enhancement solution 20 (PerkinElmer, 4001-0010) was used to activate europium fluorescence. Intensity was measured using a Victor Nivo Multimode plate reader (excitation at 320 nm and emission at 615 nm) and related to the negative control. Decreased fluorescence relative to the negative (CTXB-only) control equalled V$_H$H-blocking capacity of CTXB–GM1 interaction.

Similar to the selection of monovalent V$_H$Hs in *E. coli* cultures, purified monovalent V$_H$Hs and the generated bivalent V$_H$H BL3.2 were evaluated in a GM1 receptor-blocking DELFIA. Biotinylated CTXB binding to GM1 was analysed through pre-incubation in the absence or presence of V$_H$Hs at a 10:1 or 1:1 molar ratio (V$_H$H:CTXB) at 37 °C for 30 min before addition to the GM1-plate. Intensity measurements were normalized against a control mixture, containing CTXB and a V$_H$H construct without CTXB-specificity at the same ratio.

## Sequencing and identification of unique monovalent V$_H$Hs

The **pSANG10-3F** vector of *E. coli* clones expressing selected V$_H$Hs (CTXB–GM1 blocking capacity >25%) was sequenced (Eurofins Genomic, Germany) using primer pBDS100-1 (GTATGTTGTGTGGAATTGT-GAGC), and their CDR3s were compared to identify unique binders.

## Plasmid construction and V$_H$H dimerization

Bivalent V$_H$H constructs were created by genetically joining two identical V$_H$Hs via a Gly–Ser (G$_4$S)$_3$ linker. For in vitro evaluation, binder selection, and human cell studies, bivalent V$_H$H constructs were synthesized by GenScript (Netherlands) and cloned into the *E.coli* expression vector **pSANG10-3F** using *NotI* (New England Biolabs,

R3189S) and *NcoI* (New England Biolabs, R3193L) restriction enzymes. For the remainder of the experimental data generated, including the production of BL3.1 paratope variants, V$_H$Hs were produced using a *Komagataella phaffii* expression system. The *K. phaffi* strain CBS2612 (CBS-KNAW culture collection) was transformed with an integrative plasmid harbouring the Zeocin resistance marker, and the V$_H$H gene expressed using the methanol-inducible AOX promoter and the α-mating factor secretion signal[67]. The previously characterized anti-CTX V$_H$H was synthesized by GenScript (Netherlands) based on available sequence information and produced using *K. phaffi*[29,42]. For affinity determination, a tag-free (His- and FLAG-tag) version of the monovalent BL3.1 was expressed using the *K. phaffi* expression system, with the constitutive GAP promoter[68].

## Expression of monovalent and bivalent V$_H$H constructs

V$_H$H constructs cloned into the **pSANG10-3F** vector and transformed into *E. coli* BL21 (DE3) were incubated shaking (220 rpm) overnight in lysogeny broth supplemented with 50 µg ml$^{-1}$ kanamycin at 37 °C. From the overnight culture, 100 µl was used to inoculate 100 ml of autoinduction medium incubated for 24 hours at 25 °C while shaking at 170 rpm. Cells were centrifuged at 4300 g for 15 min and the supernatant discarded, before re-suspension in 5 ml g$^{-1}$ of cell pellet in TES (30 mM Tris–HCl pH 8.0, 1 mM EDTA, 20% sucrose (w/v)) supplemented with cOmplete Protease Inhibitor Cocktail (Roche, 1 tablet per 50 ml), 1.5 mg g$^{-1}$ of cell pellet lysozyme (Sigma-Aldrich, 62971), and 100 U DNase 1 (AppliChem, A3778) per 10 ml of TES. Following incubation on ice, the cell solutions were centrifuged at 15,000 g for 15 min. Supernatants were transferred to separate tubes and cells were resuspended in 5 ml of 5 mM MgSO$_4$ per gram of cell pellet, supplemented with identical concentrations of cOmplete Protease Inhibitor Cocktail, lysozyme, and DNase 1 as the TES buffer described above. Re-suspended cells were further incubated for 20 min on ice, centrifuged at 15,000 g for 15 min, and the supernatants were pooled with the supernatants obtained from the previous step. Pooled supernatants were centrifuged at 20,000 g for 40 min to yield the final periplasmic lysate.

The monovalent BL3.1 and the bivalent BL3.2 expressed by *K. phaffii*, were produced in 1 L Tunair Shake Flasks (Sigma-Aldrich). At first, 100 ml of overnight culture (BMGY media) was diluted to OD$_{600}$ = 1 in either 1 l BMY media supplemented with 2% glucose (BL3.1) or 1 l BMMY media (BL3.2) and kept shaking overnight at 30 °C and 160 rpm. The cultures were supplemented daily with either 1% glucose (BL3.1) or 1% methanol (BL3.2) for 2 days. At the end of the expression campaign, the media was centrifuged at 15,000 g for 15 min, and the supernatant collected for purification.

## Purification of V$_H$H constructs

Purification of V$_H$H constructs from *E. coli* periplasmic lysate was carried out through immobilized metal affinity chromatography (IMAC) using gravity flow columns. At first, 2 ml of HisPur Ni-NTA resin (Thermo Scientific, 10038124) was added to chromatography columns (Bio-Rad, 7321010). The columns were washed twice with IMAC wash buffer (PBS with 200 mM NaCl and 20 mM imidazole), before the periplasmic lysate was added. The washing procedure was repeated, IMAC elution buffer (PBS with 200 mM NaCl and 250 mM imidazole) was added to the column, and the V$_H$Hs were collected. Samples were dialyzed against PBS using dialysis tubing with a 3.5 kDa MWCO (Thermo Scientific, 68100). Dialyzed V$_H$Hs were loaded onto a 120 ml HiLoad 16/600 Superdex 75 pg column connected to an NGC purification system (Bio-Rad) for SEC and eluted into PBS. Protein concentration was determined by spectrophotometer (Thermo Scientific, NanoDrop One) at 280 nm.

For purification of the tag-free monovalent BL3.1 used for affinity studies, the *K. phaffi* supernatant was concentrated against PBS using the 10 kDa MWCO Vivaflow 200 Crossflow Cassette (Sartorius). The

concentrate was loaded onto a 120 ml HiLoad 16/600 Superdex 75 pg column connected to an NGC purification system (Bio-Rad) for SEC and eluted into PBS. The protein concentration was determined by spectrophotometer (NanoDrop One, Thermo Scientific) at 280 nm or using a BCA Protein Assay (Thermo Scientific, 23225).

Similarly, purification of the bivalent $V_HH$ construct BL3.2 from the supernatant of *K. phaffii* was also conducted using IMAC reagents. At first, imidazole (20 mM) was added to the supernatant and the pH adjusted to 7.5. Additionally, a 50 ml 1:1 mixture (w:w) of HisPur Ni-NTA resin and IMAC wash buffer was added, prior to incubation (30 min) at room temperature and, later, vacuum filtration with a bottle-top filter. The filter was washed three times with IMAC wash buffer and then eluted in two subsequent steps: 20 ml IMAC elution buffer in the first step; and 15 ml 1 M imidazole in the second step.

### Simulated gastrointestinal stability assay
The stability of BL3.2 at pH conditions representative of GI passage was assessed by incubating BL3.2 in either SGF (pH 1.2) or SIF (pH 6.8) and afterward determining CTXB-binding capacity via a DELFIA assay. SGF was prepared to a final concentration of 35 mM NaCl adjusted to a pH of 1.2 using HCl. SIF was prepared to a final concentration of 50 mM $K_3PO_4$ adjusted to a pH of 6.8 using NaOH. GF and SIF were pre-warmed together with PBS (pH 7.4) at 37 °C for 15 min. BL3.2 was mixed with either SGF, SIF, or PBS (untreated control) to a final concentration of 100 µg ml$^{-1}$ and incubated at 37 °C. At each time point, a sample was taken by removing 200 µl and adding 100 µl of $Na_2CO_3$ (SGF) or PBS (SIF and PBS) and stored at −20 °C until analysis. In parallel, a sample of BL3.2 in PBS (100 µg ml$^{-1}$) was stored in the fridge at 4 °C as a control for the full length of the experiment (5 h). The experiment was repeated twice for all conditions, and samples analysed in duplicates.

### Thermal stability assay
The thermal stability of BL3.2 and CTX was determined by a real-time PCR (QuantStudio 6 Pro) protein melt assay (Protein Thermal Shift™) using a fluorescent dye specific to hydrophobic regions of proteins (Applied Biosystems, 4462263). In a total reaction volume of 22.5 µl, 2.5 µl of fluorescent dye (8X) was added to 10 µg of either BL3.2 or CTX in PBS (pH 7.4). The thermal profile was set to a first step of 20 °C and a second step of 99 °C, both for a duration of 2 min, with a ramp rate of 0.05 °C s$^{-1}$. The Protein Thermal Shift Software (version 1.4) from Applied Biosystems was used to calculate the Derivative curve determined $T_m$.

### Affinity determination
The kinetic parameters of the monovalent $V_HH$ BL3.1 were determined using both BLI and SPR. For BLI measurements, an Octet RED96 (ForteBio) equipped with a streptavidin biosensor (Satorius, 18-5020) was used to capture 76 nM of biotinylated BL3.1 to a spectral shift of 0.8 nm. BL3.1-bound biosensors were subjected to a 4-fold serial dilution (120–0.470 nM) of CTXB in running buffer (10 mM HEPES, 150 mM NaCl, 3 mM EDTA, 50 mM MES hydrate, 0.05% P20, pH 7.24) and a control without CTXB. CTXB association and dissociation were measured for 600 seconds each. The control biosensor signal was subtracted, the data fitted with a global model (1:1 binding sites), and kinetic parameters calculated using Octet Analysis Studio v12.2.2.26 (ForteBio).

For SPR analysis, CTX was immobilized as the ligand (5 µg ml$^{-1}$ at pH 5) on a CM5 sensor chip (Cytiva, 29149603), and the kinetic interaction with BL3.1 was evaluated using a Biacore 8 K+ GoldSeal (GE Healthcare). BL3.1 was diluted in HBS-EP+ buffer (Cytiva, BR100669) and analysed at several different concentrations (1–500 nM) using single-cycle kinetics. The startup phase was set to 180 s of contact time and 300 s dissociation time, at a flow rate of 30 µl min$^{-1}$ (25 °C). The analysis phase of BL3.1 was set to a contact time of 180 s and a dissociation time of 500 s, at an identical flow rate. Results were evaluated

by subtracting the reference flow cell and HBS-EP+ buffer signals from sample data and fitting the data (excluding refractive index and baseline drift) to a global model (1:1 binding). Kinetic parameters were determined using the Biacore Insight Evaluation Software v5.0.18.22102 (Cytiva).

### Bioluminescent detection of cAMP in HCA-7 cells
HCA-7 cells (AddexBio, C0009003) were grown in white poly-D-lysine-coated 96-well plates (Corning, 354651) at 37 °C (5% $CO_2$ and 95% air) using Dulbecco's modified Eagle's medium (Thermo Scientific, 10722804) supplemented with 10% fetal bovine serum (Thermo Scientific, 12309852) and 1% penicillin-streptomycin (Thermo Scientific, 11556461). Cells were seeded at a density of $1 \times 10^4$ cells in each well. Cell culture media was removed and various dilutions of BL3.2 (0.240–31.25 nM) were incubated with a fixed concentration (0.115 nM) of CTX in complete induction buffer (Promega) at 37 °C for 30 min, before 40 µl of the mixture was added per well. A monovalent anti-CTX $V_HH$ and a bivalent $V_HH$ construct without specificity for CTX were included for comparison.

After 2 h, the cAMP detection solution (10 µl per well) and Kinase Glo-Reagent (50 µl per well) were added according to the manufacturer's protocol (cAMP-Glo™ Max Assay, Promega, Madison, WI). Intracellular cAMP was measured at 700 nm on a Victor Nivo Multimode plate reader. The effect of BL3.2 and the negative bivalent $V_HH$ control on intracellular levels of cAMP was analysed in biological duplicates, and each dilution measured in technical triplicates. The anti-CTX $V_HH$ was analysed once, with each dilution measured in technical triplicates. The cAMP standard curve was measured in triplicates and levels of intracellular cAMP interpolated using a sigmoidal four parameter logistic regression model ($R^2 = 0.9543$) in GraphPad Prism version 9.5.0. Relative $IC_{50}$ for BL3.2 was determined using a variable slope model (GraphPad Prism version 9.5.0).

### In silico prediction of BL3.2-CTX interfaces
ColabFold in combination with classical molecular dynamics simulations was used to predict and characterize the protein–protein interface of BL3.2 with CTXB. Four distinct binding poses between BL3.2 and CTXB were identified and further validated (two repetitions of 1 µs each) by classical molecular dynamics simulations using the AMBER 22 simulation software package which contains the pmemd.cuda module[69]. The structure models were placed into cubic water boxes of TIP3P water molecules with a minimum wall distance to the protein of 10 Å[70–72]. Parameters for all simulations were derived from the AMBER force field 14SB[73,74]. To neutralize the charges, uniform background charges were used[75–77]. Each system was carefully equilibrated using a multistep equilibration protocol[78].

Bonds involving hydrogen atoms were restrained using the SHAKE algorithm, allowing a timestep of 2.0 fs[79]. The systems pressure was maintained at one bar by applying weak coupling to an external bath using the Berendsen algorithm[80]. The Langevin Thermostat was utilized to keep the temperature at 300 K during the simulations[81]. A cluster analysis was carried out in AMBER's CPPTRAJ program using the same RMSD distance cut-off criterion of 5 Å for all trajectories[82]. Protein–protein contacts were also quantified using the GetContacts software (Stanford University)[83] and interaction energies calculated using the LIE tool implemented in CPPTRAJ[82].

The BL3.2–CTX interface was also independently identified and characterized by Raven Biosciences (Denmark) and their proprietary EpiC platform v.9. The structure of BL3.2 was built from its amino acid sequence using NanoBodyBuilder2[84]. The structure of CTXB was based on an apo X-ray crystal structure (PDB ID: 1XTC)[85]. The homopentamer with chain identifiers A–E were kept, and missing atoms and residues were added using PDBfixer. The EpiC platform generated a structural ensemble of CTX (50 distinct conformations) before performing extensive docking and molecular dynamics simulations to identify the

$V_H$H–antigen interface in atomistic resolution. The crude CTXB epitope was defined as the toxin consensus residues observed to interact directly with glycolipid receptors in experimentally determined structures, specifically toxin residues 11, 12, 13, 33, 51, 56, 57, 58, 61, 88, and 91[5,86]. The EpiC platform ensured that the $V_H$H was interacting with at least a subset of these residues in the initial docking step. The $V_H$H was allowed to refine its binding interface freely in the latter docking and molecular dynamics steps.

## Protein complex separation by SEC

Complex formation between BL3.1 or BL3.2 with CTXB was analysed by SEC on an ÄKTA system (GE Healthcare). Recombinant CTXB (52 μM) was mixed with BL3.1 (104 μM) or with BL3.2 (52 μM). These mixtures were incubated for 20 min at room temperature, and subsequently loaded onto a Superdex 200 Increase 10/300 GL column (Cytiva) equilibrated with PBS (pH 7.4). Chromatograms were normalised to the maximum peak absorbance and compared to independent SEC runs with the individual $V_H$H constructs.

## HDX–MS analysis of BL3.1–CTXB interaction

The exchange reaction was started by diluting either CTXB (50 nM) or the CTXB–BL3.1 complex (50 nM CTXB and 250 nM BL3.1) in a deuterated buffer (150 mM NaCl, 2.7 mM KCl, 25 mM HEPES at pH 7.4). The final $D_2O$ concentration in the exchange reaction was 95%. The reaction was quenched at different time points (1, 10, 40, and 100 min) by mixing equal volumes of the reaction liquid and ice-cold quench buffer (1% formic acid, 0.1 M TCEP, trifluoroacetic acid 0.025%, and 1 M guanidine hydrochloride). The pH of the quenched reaction was 2.5. Following quenching, the reaction was frozen in liquid nitrogen and stored at −80 °C. Each time point of the experiment was conducted with three technical replicates.

The quenched samples were injected on a nanoACQUITY UPLC system with HDX technology (Waters Corporation) and passed through a 2.1 × 30 mm pepsin column with POROS 20 AL resin (Thermo Scientific, 1602906). The proteolyzed sample was immediately directed to a 2.1 × 5 mm trap column (Waters Corporation, 1.7 μm Acquity Vanguard BEH C18) to desalt peptides. The flow rate was set to 70 μl min⁻¹ during 4 min of trapping with buffer A (0.2% formic acid, 0.01% trifluoroacetic acid at pH 2.5). After desalting, the peptides were separated with a 1.0 × 100 mm analytical column (Waters Corporation, 1.7 μm Acquity Vanguard BEH C18) with a linear 5–50% acetonitrile gradient using buffer B (99.9% acetonitrile, 0.1% formic acid, and 0.01 % trifluoroacetic acid at pH 2.5). The elution gradient was run at 40 μl min⁻¹ for 17 min. The output of the analytical column was directed to a mass spectrometer (Waters Corporation, Q-TOF SYNAPT G2-Si) for peptide identification and determination of the deuterium uptake. The mass spectrometer was operated in the positive ion electrospray mode, with the ion mobility function to minimize spectral overlap using the MSE acquisition mode (Waters Corporation). Lock mass correction with the Leu-ENK peptide was used to ensure mass accuracy.

A library of non-deuterated peptides was created using the ProteinLynx Global server 3.0 (PLGS) (Waters Corporation) using the following requirements: (1) A mass error for the peptide must be below 10 ppm for the precursor ion, (2) the peptide must have at least two fragmentation products, and (3) the peptide must be identified in at least three out of six non-deuterated runs. The level of deuteration in the peptides was determined with DynamX 3.0 (Waters Corporation). The difference in deuteration ($\Delta D$) between two states (CTXB, and CTXB–BL3.1, respectively) was calculated by normalization with respect to the theoretical maximum uptake (MaxUptake = $N - P - 2$), where N is the number of amino acids in the peptide and P is the number of prolines). The percentage of deuteration was determined according to: $\Delta D(\%) = \frac{\Delta D}{Max\ uptake} \times 100$.

## Bacterial growth conditions

A streptomycin-resistant clinical isolate of *V. cholerae* (HaitiWT), isolated during the 2022 outbreak, was used in this study[87]. *V. cholerae* was grown with aeration (300 rpm) in LB (Fisher Scientific, BP1426-500) supplemented with streptomycin (100 μg ml⁻¹), at 37 °C for 16–18 h.

## BL3.2 inhibition of CTX activity in vivo

The inhibitory effect of BL3.2 on net fluid secretion in the SI, diarrhoeal onset, and bacterial colonization in the SI following intragastric administration of CTX or overnight culture of a live *V. cholerae* strain was investigated in a previously described 5-day-old CD-1 mouse model[35]. In essence, fluid accumulation and weight loss were measured for all mice exposed to CTX, and, bacterial colonization was determined for mice inoculated with *V. cholerae*.

In the first experiment, BL3.2 (9 mg ml⁻¹) was pre-incubated at 37 °C for 30 min with CTX (1 mg ml⁻¹) before a single (50 μl) oral gavage of the mixture was administered. For the subsequent experiments, three administrations (50 μl each) were given to each mouse through oral gavage: a first administration of either the bivalent $V_H$H construct (9 mg ml⁻¹, in PBS) or 9 mg ml⁻¹ BSA (Sigma-Aldrich, A9418) in PBS as a control; a second administration of either CTX (0.1 mg ml⁻¹) or live *V. cholerae* (2.8 × 10⁸ CFU), and a third administration of either the bivalent $V_H$H construct (9 mg ml⁻¹, in PBS) or BSA (9 mg ml⁻¹, in PBS) as a control. This infant mouse model is limited to a maximum of three oral gavages, due to the physical stress sustained by the animal. Food colouring (5 μl ml⁻¹) was added to all oral mixtures prior to administration, to be able to confirm correct execution of oral administration and, later, detect diarrhoea. Mice were weighed after the final oral administration. For mice infected with *V. cholerae*, a serial dilution of the inoculum was plated on agar plates supplemented with streptomycin to determine bacterial input dose.

Mice were monitored every hour during the experiment, and the experiments terminated due to the moribund state of the mice in the control group. Mice were weighed, and euthanized, and their entire SI extracted. Weight loss was determined by comparing the weight of each mouse after the final intragastric administration with the weight prior to euthanasia. The excised SI was used to determine the fluid accumulation ratio as described elsewhere[88]. CFU in the SI were determined by homogenizing the SI using metal beads and a Mini-Beadbeater-24 (Glen Mills) followed by serial 10-fold dilutions and plating on agar plates supplemented with streptomycin.

## Data processing and visualization

GraphPad Prism version 9.5.0 was used for figure generation and all statistical analyses. Data were analysed using the Mann–Whitney U test, in which differences were considered significant (*) at P values of ≤ 0.05 and highly significant (**) at P values of ≤ 0.01. Average values and standard deviations were calculated after transforming the values to the figure scale illustrated. CLC Main Workbench version 23.0.2 was used for sequence analysis and alignment.

## Reporting summary

Further information on research design is available in the Nature Portfolio Reporting Summary linked to this article.

# Data availability

The BL3.1 and BL3.2 protein sequence data used in this study are available in the Mendeley database (https://data.mendeley.com/datasets/thvh9j7hbk/1). The mass spectrometry proteomics data generated in this study have been deposited to the ProteomeXchange Consortium via the PRIDE partner repository at https://www.ebi.ac.uk/pride/archive/projects/PXD057713[89]. Source data are provided with this paper.

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

## Acknowledgements

We thank Innovation Fund Denmark (grant 1044-00106B) for their financial support to M.P. F.G.Z. and M.K.W. are supported by NIH (R01AI042347) and HHMI. The research by F.G.Z. was also supported by the Life Sciences Research Foundation (Zingl-2024HHMI). D.S.P. and N.S. are supported by NCMM and the Research Council of Norway (grants 187615 and 325528). We would also like to thank Anna Engel for her contributions to in vitro evaluation of $V_H$H constructs and Nick J. Burlet for his technical assistance with the BLI assay. We acknowledge EuroHPC Joint Undertaking for awarding us access to MeluXina, Luxembourg. Schematic illustrations were created using BioRender.com.

## Author contributions

M.P., S.W.T., A.H.L. and L.G. conceptualized the project. E.R.-R. and E.W.A. generated the $V_H$H libraries and performed initial in vitro screenings. E.R.-R. contributed to data interpretation throughout the project. J.K.H.R. designed plasmid constructs optimized for protein expression. H.T.H. assisted with cloning and transformation. M.K.W. and F.G.Z. designed the animal studies. M.P. and F.G.Z. executed the animal studies. T.P.J. and M.L.F.-Q. performed and analysed the in silico epitope predictions. U.K. coordinated experimental validation of the protein structure analysis. N.M. performed and analysed size-exclusion chromatography studies and assisted in HDX-MS studies. D.S.P. and N.S. designed, executed, and analysed HDX-MS experiments. M.P. and F.G.Z. analysed the data and drafted the manuscript. All authors were a part of the manuscript process and final review.

## Competing interests

All authors affiliated with Bactolife A/S are present or past employees of Bactolife A/S. A.H.L. and S.W.T. are shareholders of Bactolife A/S. The remaining authors declare no competing interests.
