## [Transparent Peer Review file · Nature Communications]

Orally delivered toxin-binding protein protects against diarrhoea in a murine cholera model

Corresponding Author: Dr Sandra Wingaard Thrane

Version 0:

Reviewer comments:

Reviewer #1

(Remarks to the Author)

Petersson et al describe the isolation of new cholera toxin B-subunit specific VHHs and show oral delivery of a bivalent VHH-VHH molecule to infant mice reduces intestinal fluid secretion, diarrhea and bacterial load in the small intestine. The work is novel in that few (if any) orally delivered CTX-specific antibodies have been tested for the treatment of *V. cholerae*. Experiments are logically presented; however, a number of concerns and suggestions are raised below that should be addressed before considering publication in this journal

1. In vitro and in vivo studies would have benefited from the inclusion of anti-CTX mAb controls as benchmarks. The authors identify several manuscripts reporting these mAbs (i.e., Verjan Garcia et al Sci Rep 2023). As it stands, there is no comparator in any of the studies shown
2. The BL3.1 monomer VHH was not directly tested head-to-head vs the BL3.2 dimer in Figure 1. It is implied, but it would be important to show the results of 1:1 VHH:CTXB for BL3.1 in Figure 1b. Additionally, an irrelevant VHH-VHH (like that used in Fig 3) should be shown in Figure 1b as a control. Were bivalent VHH-VHHs tested from the other 4 VHHs in Fig 1a, since it's conceivable even the less potent monomer inhibitors could be superior as dimers? Were any biparatopic VHH-VHHs tested?
3. Figure 3 would have benefited from a benchmark mAb control. Is the newly isolated VHH-VHH far superior?
4. The affinities and kinetics being reported are apparent binding affinities. The BLI experiment immobilized VHH-VHH and flowed CTXB, which is a pentamer. This will result in a significant amount of avid binding. To collect monovalent KDs the CTXB should be immobilized and VHH flowed as analyte. Replicates should be reported.
5. The structural modelling in Fig 4 does not add to the manuscript without confirmatory studies (ie., Xray co-crystal structures/cryoEM/HDX-MS). It should be moved to supplemental. Overall, the manuscript could have benefited from a deeper interrogation of the bivalent VHH-VHH neutralization mechanism and stoichiometry (ie, does the VHH-VHH bind within the same CTXB pentamer, or is it capable of binding separate toxin molecules, or are both modes possible?)
6. For the animal studies in Fig 5 and 6, is there a reason why the infant mouse model was used over the infant rabbit model (used by one of the co-authors in other high impact papers)? How were the dosing / reporting times chosen? Was the VHH-VHH concentration in the SI ever tracked or investigated? The animal studies would have benefited by dosing an irrelevant VHH-VHH control antibody as well.
7. There is no attempt to examine formulation in this work (see manuscripts from VHSquared and their oral anti-TNFa work). Would co-administering with something like BSA have increased the potency/effects?
8. More discussion around the practicality of this oral VHH-VHH approach (dose requirements, dosing regime and how it was chosen, cost, production) in the Discussion is needed.
9. Authors are encouraged to test their working hypothesis of how anti-toxin VHHs reduce the bacterial burden in the SI. The paragraph in the Discussion around this topic raises some interesting points – testing this would increase the impact of this

work.

Reviewer #2

(Remarks to the Author)

Nice study. Well written. Important topic. Data support conclusions. Describes development, characterization and pilot preclinical evaluation of a bivalent camelid derived VHH construct expressed in yeast (BL3.2). Strengths include approach, and early screening for toxin receptor binding disruption to down select clones for subsequent development. Data convincing that a product is made and disrupts CTX to GM1 binding. Some stability data regarding in vitro conditions of wide pH c/w gastric and small intestine environments supportive. In silico analysis suggestive of potential critical amino acids in the interaction. Supportive data that BL3.2 decreases cAMP in colonic cell culture line. Supportive data in small animal studies, that BL3.2 can improve outcome in neonatal mice against diarrhea and weight loss against cholera holotoxin challenge, and against diarrhea, weight loss and CFU against virulent *V. cholerae* challenge.

Abstract says "great promise": please remove adjective

Abstract says "very low cost" but no cost data provided; please remove this description.

Please revise abstract to summarize only the primary data in the manuscript.

Are the amino acids involved in CtxB predicted to be displayed when CtxB forms its heterodimer CTA and pentameric CTXB structure as cholera holotoxin? Does it impede the ability of the CtxB pentamer to form, disassociate or bind to GM1. In silico seems to predict the latter, but comments on others should be made.

A weakness that should be pointed out is that no mutational studies were performed to confirm purported critical amino acid interactions.

Are the amino acids predicted by in silico conserved in LT ETEC or just classical and El tor CT? Please mention/discuss.

No data are provided with regard to how much product would be required to coat intestine and how long it is present in intestines.

Many undigested food products pass through intestines in 18-24 hours in humans.

Would mention that no such data yet checked for this product, and as such unclear how often and how much of a quantity would need to be ingested to be clinically impactful.

The authors suggest "could be dietary supplement" so should comment on this limitation since minimal data provided in this regard.

Authors rightly point out breast milk is protective, but babies suckle many times during the day, and probably require far less antibody since they are not ingesting other things contaminated with *V. cholerae*

These limitations should be discussed.

68% blocking, and 27 fold decrease cAMP hard to put into context.

Appears that 25-50 mcg of CT administered to mice per my calculations

That is a large load that often kills neonatal mice within 18-48 hours.

Why did no mice die in the BSA control group??

If desire was to harvest/sacrifice before death then would state, but surprising no death at 22 hours.

Were control mice moribund at that stage?

Why was BSA used as the control and not a non-specific bivalent camelid VHH in the animal studies.

Would mention as limitation/weakness.

The animal studies are supportive but cohorts of animals are small (5 or 6).

Only one animal model used; would mention as limitation; for instance could have evaluated direct challenge in rabbit ileal loops.

Mention should be made that a product like VHH may have its best utility as a prophylactic for individuals at immediate risk of cholera, and could supplement control programs for such populations such as distributing chlorine tablets (also a very cheap intervention for high risk populations). Mention should be made that CT is a very potent enterotoxin and once internalized by an intestinal epithelial cell that the window of any protection afforded by a VHH to that cell has passed; as such, one might expect a limited role in therapy of cholera patients as opposed to preventing disease before it occurs) although no data currently exist (would explicitly state to assist reader).

Overall, well written study whose central conclusions regarding initial development and evaluation of a product that disrupts CT-GM1 interaction are supported by the data provided.

Reviewer #3

(Remarks to the Author)

The authors generate and characterize an orally deliverable bivalent VHH construct (BL3.2) that binds to the B-subunit of cholera toxin (CTX). This VHH construct interferes with interaction between CTX and receptor GM1, which is critical for the gastrointestinal pathogen *V. cholerae*. The authors show the engineered bivalent is stable under physiological gastrointestinal passage via in vitro studies, as well as active in functional mouse studies. Such a reagent is low cost with potential high payoff in terms of potential impact on populations suffering from cholera. The paper should be of interest to those in the field, as well as the general audience. The manuscript is well-written with clear data/figure presentations.

Comments/suggestions:

1. Not clear why authors are not seeing near 100% blocking, based on assessment of high affinity of BL3.1 and BL3.2. What's known about the affinity of the CTX/GM1 interactions? It would be beneficial for a sentence or two to be included.
2. Significant figures, such as extended data table 1, should be reported to the highest least significant digit. For instance kon should be 3.69 +/- 0.03 (i.e., the part in parenthesis of the error is not relevant 3."11")
3. Methods section: provide methods on how VHH were produced for screening after subcloning into pSANG10 vector.(20) It is not apparent how authors performed expression/purification of 380 monovalent VHH constructs. In addition, reference 20 seems to be an intermediate reference and actual reference to pSANG is C.D. Martin, G. Rojas, J.N. Mitchell, K.J. Vincent, J. Wu, J. McCafferty, D.J. Schofield. A simple vector system to improve performance and utilisation of recombinant antibodies ?? Reading results/methods, it sounds like the VHH clones were expressed to the supernatant/culture media, which doesn't seem to be the case. Details should be present in the methods.
4. DELFIA- provide full name before abbreviation
5. Methods: thermal stability experiments: conditions not clear e.g., buffer, pH?
6. Thermal stability experiments suggested the bivalent BL3.2 possesses a significantly higher melting point (more than 15 degree C) than the single VHH. If true, this would suggest some sort of VHH:VHH interaction. However, it is more likely there may be something up with the analysis or data. As the raw data are not included, I cannot comment. Raw data should be included in supplemental, as well as something describing what the authors believe is going on. If true, it would be of interest.
7. SPR data analysis of BL3.1: thank you for including data/fits/residuals. Model/Data not the best fit, but consistent with what appears to be significantly high affinity (near low nM). Suggestion for the future....perform such concentration studies with replicates.
8. Authors state: " BL3.2 achieved half-maximal relative inhibitory concentration (IC50) at 1.5 nM, corresponding to a VHH-VHH:CTX molar ratio of 13:1." Seems like a strange way to compare molar ratios, as VHH-VHH is a dimer and CTX is a pentamer (based on likely binding site). This would work out to a 6.5 to 5 or roughly 1:1 VHH to CTXB ratio.
9. Binding site predictions: authors state "Molecular dynamics simulations of Model 4 defined the epitope-paratope interaction at high resolution." The "at high resolution" sounds really strange for a computational model, as all models will provide high resolution (x,y,z coordinates). Please remove high resolution.

Version 1:

Reviewer comments:

Reviewer #1

(Remarks to the Author)

Thank you for incorporating many of the suggested changes, including significant improvements to the mechanistic understanding via HDX-MS data and generation of mutants. I am still of the opinion a control VHH-VHH and/or the anti-CTX VHH from Goldman et al 2006 (formatted as a VHH-VHH) would have been preferred to BSA in the in vivo animal experiments; however, I accept the arguments put forth in the rebuttal.

Minor comments:

Line 97-98: please change "...with a KD determined to be 0.76 nM.." to "...with an apparent KD determined to be 0.76 nM...". Also change "...and to be 85.5 nM..." to "...and a monovalent KD of 85.5 nM..." to reflect the avid binding event occurring in the first assay orientation. It is also fine to omit the word "monovalent", if the authors prefer, when referring to the second affinity value

Supp Fig 10: the complex formation by SEC is helpful, although SEC-MALS would have been better suited. Can SEC MW standards be labelled on the chromatogram in an attempt to guide the reader? (I recognize this isn't analytical SEC, but even some reference point with MW stds will help with interpretation of peaks/complexes).

Editorial note: this reviewer was also asked to comment in place of reviewer 2 who was unable to provide a response at this time:

I have reviewed the responses to R2 and believe all concerns have been adequately addressed.

Minor comments to Authors:

1. Sentence on lines 241-243 needs to be edited. As written, it suggests single-domain antibodies can be derived from IgAs
2. Please explicitly state limitations of this study in the Discussion in 1-2 sentences
 - small animal cohort for in vivo testing
 - use of BSA instead of control VHH (irrelevant or anti-CTX from Goldman) in the in vivo expts

-dosing not examined/optimized due to limitations of model

Reviewer #3

(Remarks to the Author)

The modifications addressing all reviewer comments have improved the manuscript. I am glad to see that revisiting the binding/blocking experiments identified conc. issues that when corrected, now provide a more consistent set of data. Please see a few additional comments/suggestions.

1. Methods

The authors should be more explicit in their methodology. Errors were their protein concentrations were uncovered during their revisions. How specifically were VHH concentrations determined? The authors state via 280 nm absorbance. Please state the 280 nm extinction coefficients. I'm assuming values from something like (ProtParam:

<https://web.expasy.org/protparam/> were used). This would be the most accurate and accessible method.

It also appears a BCA assay was used to determine concentration for affinity measurements. Any particular reason why the BCA assay is being used (which is more appropriate for total heterogeneous protein concentration) rather than direct (more accurate) 280 extinction coefficient for this quantitative measurements? I would not be surprised if differences in conc. were stemming from BCA from one time/individual to another.

2. Results: line 89 "BL3.1 displayed high affinity for CTXB, with a KD determined to be 0.76 nM by BLI (with BL3.1 as ligand) and to be 85.50 nM by SPR (with CTXB as ligand) (Supplementary Table 1). These values are similar to the KD (77 nM) of a previously reported anti-CTX VHH 30."

- This is a 100-fold difference. Since this is almost certainly due to avidity effects with the way the authors setup the initial BLI experiments, they should be direct and state this well-known issue for the likely difference in K values.

3. Thermal denaturation experiments: Thank you for the clarification and including the raw data in the Suppl. Info. Based on the authors assessment: "Protein thermal stability screening using differential scanning fluorimetry (Protein Thermal Shift™) showed that BL3.2 had even greater thermal stability (67.3 °C) than CTX (52.0 °C) (Fig. 2b and Supplementary Fig. 3)"

- Examining the raw differential scanning fluorimetry data, the T_m of 67.3 for BL3.2 looks accurate. This is not the case for the CTX runs. There is just not enough of a change in signal between native to say the T_m is 52. Almost appears if T_m was identified without looking at data, but derivative signal alone. If anything, based on the general expected trend for DSF data, there is a small increase in signal right before the max raw signal of ~72dC, which would suggest the mid-point is very close to ~67dC, suggesting almost identical melting temperatures between the two VHH constructs. At a minimum, I'd suggest using the max derivative peaks which occur right around 67dC. Even better, signal response can be optimized by varying protein concentration and dye to get the best signal change to increase confidence in the observed T_m value.

4. Suppl Figure 10: appears to be missing the free BL3.1 and BL3.2 species profile to aid interpretation.

Reviewer #1

Petersson et al describe the isolation of new cholera toxin B-subunit specific VHHs and show oral delivery of a bivalent VHH-VHH molecule to infant mice reduces intestinal fluid secretion, diarrhea and bacterial load in the small intestine. The work is novel in that few (if any) orally delivered CTX-specific antibodies have been tested for the treatment of *V. cholerae*. Experiments are logically presented; however, a number of concerns and suggestions are raised below that should be addressed before considering publication in this journal.

Response:

Thank you for the positive comments and constructive critique. Below we address the issues raised and implemented the reviewers' suggestions to the best of our ability.

- 1. In vitro and in vivo studies would have benefited from the inclusion of anti-CTX mAb controls as benchmarks. The authors identify several manuscripts reporting these mAbs (i.e., Verjan Garcia et al Sci Rep 2023). As it stands, there is no comparator in any of the studies shown.**

Response:

We agree that including an anti-CTX control would be good. However, in our view, using in vivo mAbs against CTX (i.e., Verjan Garcia et al Sci Rep 2023) are less relevant controls due to the extensive degradation that such mAbs undergo in the GI tract and, thus, limited application for neutralization of pathogens in the small intestine¹. Therefore, we now include a comparison against the anti-CTX V_HH construct by Goldman et al which was previously developed as a tool for cholera detection². We now emphasize the benefit of a V_HH format in contrast to a conventional antibody format for oral applications on line 63.

For the in vitro studies of CTXB-GM1 blocking capacity, we have now included a comparison between BL3.1/BL3.2 and the anti-CTX V_HH control (Goldman et al., 2006) as well as a negative V_HH control without CTX-specificity in Figure 1b. Similarly, we performed a comparison of their ability to neutralize the function of CTX using the HCA-7 cell assay (line 116 and Supplementary Fig. 4). These results show the stark contrast between a V_HH with CTXB specificity (e.g., by Goldman et al., 2006) and a V_HH construct (i.e., BL3.1/BL3.2) able to both bind and neutralize its receptor-binding properties. The method section has been updated on line 361, 462, and 469.

Furthermore, the previously presented BLI affinity data has now been complemented with SPR measurements to enable a direct comparison with the previously reported anti-CTX V_HH affinity by Goldman et al². This data can be found in Supplementary Table 1 and the corresponding sensorgrams added to Supplementary Fig. 2. The new data has been referenced on line 87 and the method section updated, starting on line 446.

Given the poor toxin neutralizing ability of the only other reported V_HH construct against CTXB in vitro, the limited use for mAbs against gastrointestinal pathogens, and ethical guidelines for the use of animals in research, we argue that including them as benchmarks in the in vivo studies is not relevant in this context.

- 2. The BL3.1 monomer VHH was not directly tested head-to-head vs the BL3.2 dimer in Figure 1. It is implied, but it would be important to show the results of 1:1 VHH:CTXB for BL3.1 in Figure 1b. Additionally, an irrelevant VHH-VHH (like that used in Fig 3) should be shown in Figure 1b as a control.**

Response:

Figure 1 has been updated to include a direct comparison between BL3.1 and BL3.2 at a V_HH construct: toxin molar ratio range of 1:1-100:1. Figure 1 now also contains the anti-CTX V_HH (Goldman et al., 2006) mentioned above and a V_HH control without specificity for CTXB as controls².

- 3. Were bivalent VHH-VHHs tested from the other 4 VHHs in Fig 1a, since it's conceivable even the less potent monomer inhibitors could be superior as dimers? Were any biparatopic VHH-VHHs tested?**

Response:

A few of the corresponding bivalent V_HHs were generated from the presented monomeric constructs and tested in a CTXB-GM1 blocking assay. BL3.2 demonstrated superior blocking capacity compared to the other bivalent constructs. However, given the complete (100%) CTXB-GM1 blocking capacity of BL3.1 at a 10:1 molar ratio (V_HH:CTXB) and BL3.2 at a 1:1 molar ratio (V_HH-V_HH:CTXB) now demonstrated in the updated Figure 1 (and on line 84 and 95), we argue that a comparison to other bivalent constructs is no longer relevant.

The reason for an increase in blocking capacity from 88% for BL3.1 at a 10:1 molar ratio and 68% for BL3.2 at a 1:1 molar ratio (old Figure 1) to 100% (new Figure 1) is due to a previously incorrect determination of protein concentration, which we

found during our revision process. Given the superior blocking capacity displayed by BL3.1/BL3.2 in comparison to the other constructs, the data from other bivalent V_HHs are not displayed.

4. Figure 3 would have benefited from a benchmark mAb control. Is the newly isolated VHH-VHH far superior?

Response:

As described in the response to the first comment made by the reviewer, a relevant anti-CTX V_HH control has now been included as a benchmark in the HCA-7 cell assay (Supplementary Fig. 4). BL3.2 is far superior in terms of CTX neutralization (line 116).

5. The affinities and kinetics being reported are apparent binding affinities. The BLI experiment immobilized VHH-VHH and flowed CTXB, which is a pentamer. This will result in a significant amount of avid binding. To collect monovalent KDs the CTXB should be immobilized and VHH flowed as analyte. Replicates should be reported.

Response:

The reviewer raises a valid point, and we now also include SPR measurements of immobilized CTXB against flowing BL3.1. We have included the new SPR affinity data in Supplementary Table 1 and the corresponding sensorgrams for the duplicates added to Supplementary Fig. 2. BL3.1 affinity towards CTXB using both BLI (0.76 nM) and SPR (85.50 nM) is now presented on line 89.

6. The structural modelling in Fig 4 does not add to the manuscript without confirmatory studies (ie., Xray co-crystal structures/cryoEM/HDX-MS). It should be moved to supplemental. Overall, the manuscript could have benefited from a deeper interrogation of the bivalent VHH-VHH neutralization mechanism and stoichiometry (ie, does the VHH-VHH bind within the same CTXB pentamer, or is it capable of binding separate toxin molecules, or are both modes possible?)

Response:

We agree that there was a need for experimental verification of our in silico predictions. Initially, we made three attempts at co-crystallization without being successful. Now, have been able to confirm the ability of BL3.2 to interfere with the GM1 receptor binding pocket of CTX using HDX-MS studies, and these results are presented on line 139 (and onwards) and in Fig. 5. The supplementary HDX-

MS raw data for the epitope mapping has been added to Supplementary Fig. 6 and 7, and the CTX coverage map to Supplementary Fig. 9.

Furthermore, we have experimentally verified, through mutations of single amino acids, key residues in the BL3.2 paratope which align with previous in silico predictions. These results have been added to a new figure, now Supplementary Fig. 11, and addressed on line 160.

The stoichiometry between BL3.2 and CTX was also further investigated, and the results and discussion sections are now presented on line 155 (Supplementary Fig. 10). The method section has been updated, to account for CTXB production and SEC analysis (line 290 and 507) and. In short, the possibility of different protein-protein interfaces could be demonstrated in vitro by size (hydrodynamic radius) separation of various BL3.2-CTXB complexes.

7. For the animal studies in Fig 5 and 6, is there a reason why the infant mouse model was used over the infant rabbit model (used by one of the co-authors in other high impact papers)?

Response:

As emphasized by our co-author Dr. Waldor in an earlier review, the infant rabbit model is particularly suitable for obtain large amounts of in vivo derived *V. cholerae* for RNA-seq, metabolomic, and proteomic data generation³. The infant mouse model is more suitable for experiments where relatively large numbers of experimental animals are necessary. The infant mouse model selection rationale has been expanded on line 168 and reference made to previously published review by Sit et al³.

8. How were the dosing / reporting times chosen? Was the VHH-VHH concentration in the SI ever tracked or investigated? The animal studies would have benefited by dosing an irrelevant VHH-VHH control antibody as well.

Response:

Due to the physical stress sustained by the infant mice during oral gavage, the number of oral administrations was limited to three. This limitation of the model has now been emphasized to on line 567 in Methods.

To represent an intended food supplement with a potential protective effect, the first administration of BL3.2 was carried out prior to toxin administration/infection, and the second two shortly thereafter to ensure

presence of BL3.2 in the GI tract and, thus, optimize protection. This has now been detailed on line 177 and 180.

Mice were monitored every hour during the experiment and the experiments terminated due to the moribund state of the mice in the control group. This has now been added to line 575.

Given the species-dependent GI conditions (e.g., presence of degradational enzymes, pH levels, and intestinal surface exposure), translating oral dosing requirements from an animal model to intended human application comes with a high-level of uncertainty. Therefore, given the intended human application of BL3.2, we did not track the concentration of BL3.2 in the murine SI. Instead, we orally administered a theoretical excess of product needed to neutralize the CTX present in the SI of infant mice to ensure that a protective effect by BL3.2 could be observed.

For reasons detailed in the answer to the first comment by the reviewer, an additional control has not been included in the in vivo studies.

9. There is no attempt to examine formulation in this work (see manuscripts from VHSquared and their oral anti-TNF α work). Would co-administering with something like BSA have increased the potency/effects?

Response:

A discussion regarding formulation to enhance product effectivity has been added to line 259 and onwards in the discussion, with a reference to a previous study where oral rehydration salts enhanced the stability of a virus-neutralizing V_HH construct⁴. We would, however, also like to add that some V_HHs are extraordinarily stable, as we have shown in our own work, and in our experience the work done by VHSquared is in most cases not necessary⁵⁻⁷. In any case, formulation will be optimized for human applications, which are not well modelled by infant mice.

10. More discussion around the practicality of this oral VHH-VHH approach (dose requirements, dosing regime and how it was chosen, cost, production) in the Discussion is needed.

Response:

We added a discussion regarding dosing requirements in relation to previous studies of V_HH constructs targeting rotavirus and ETEC (line 257 and onwards). As explained in comment eight, the dose requirements were not investigated in our

animal experiments due to limitations of this model (now further emphasized on line 567).

11. Authors are encouraged to test their working hypothesis of how anti-toxin VHHs reduce the bacterial burden in the SI. The paragraph in the Discussion around this topic raises some interesting points – testing this would increase the impact of this work.

Response:

In this manuscript, we have demonstrated that our CTX-specific V_HH construct significantly reduces *V. cholerae* burden in the small intestine of infant mice. Previous results from in vivo studies of CTX-knockout mutants by Rivera-Chavez and Mekalanos suggest that CTX is directly involved in bacterial nutrient acquisition (i.e., long-chain fatty acids and iron)⁸. The link between this previous study and the results presented here, has now been further emphasized on line 225.

It would indeed be of considerable scientific interest to compare these findings to the predicted mechanism of action of BL3.2. However, these investigations would require extensive experimental investigation, and we consider such experiments to be beyond the scope of this work.

Reviewer #2

Nice study. Well written. Important topic. Data support conclusions. Describes development, characterization and pilot preclinical evaluation of a bivalent camelid derived VHH construct expressed in yeast (BL3.2). Strengths include approach, and early screening for toxin receptor binding disruption to down select clones for subsequent development. Data convincing that a product is made and disrupts CTX to GM1 binding. Some stability data regarding in vitro conditions of wide pH c/w gastric and small intestine environments supportive. In silico analysis suggestive of potential critical amino acids in the interaction. Supportive data that BL3.2 decreases cAMP in colonic cell culture line. Supportive data in small animal studies, that BL3.2 can improve outcome in neonatal mice against diarrhea and weight loss against cholera holotoxin challenge, and against diarrhea, weight loss and CFU against virulent *V. cholerae* challenge.

Response:

Thank you for the thorough and positive assessment of our work.

1. Abstract says “great promise”: please remove adjective

Response:

The adjective has been removed.

2. Abstract says "very low cost" but no cost data provided; please remove this description.

Response:

Very low cost has been removed from the Abstract; production costs in relation to potential dosing requirements are now addressed on line 258 and onwards in the discussion.

3. Please revise abstract to summarize only the primary data in the manuscript.

Response:

The abstract has been revised accordingly.

4. Are the amino acids involved in CtxB predicted to be displayed when CtxB forms its heterodimer CTA and pentameric CTXB structure as cholera holotoxin? Does it impede the ability of the CtxB pentamer to form, disassociate or bind to GM1. In silico seems to predict the latter, but comments on others should be made.

Response:

As illustrated by the BL3.2-CTX modelling predictions (Fig. 4) and in the comparison of machine modelling and the new HDX-MS study in Supplementary Fig. 8, BL3.2 binds to the GM1 receptor binding pocket of the CTX pentamer. Line 69 in the introduction has also been updated to emphasize that BL3.1/BL3.2 is blocking the full toxin from interacting with GM1 by specifically targeting CTXB.

5. A weakness that should be pointed out is that no mutational studies were performed to confirm purported critical amino acid interactions.

Response:

An HDX-MS analysis of BL3.2-CTX interaction, to determine amino acid interactions in the CTX epitope, has now been included (Fig. 5). Furthermore, the results from mutational studies of the BL3.2 paratope have now been included in (Supplementary Fig. 11) and addressed on line 160 and onwards.

6. Are the amino acids predicted by in silico conserved in LT ETEC or just classical and EI tor CT? Please mention/discuss.

Response:

The amino acids confirmed to be in the CTX epitope of BL3.2 are greatly conserved between hLT and CTX, there is only one amino acid difference compared to the CTX epitope now reported in Supplementary Fig. 8. The potential use of BL3.2 against other bacterial pathogens producing similar AB5-type enterotoxins (e.g., LT-producing ETEC) has now been mentioned in the final part of the discussion on line 269.

7. No data are provided with regard to how much product would be required to coat intestine and how long it is present in intestines. Many undigested food products pass through intestines in 18-24 hours in humans. Would mention that no such data yet checked for this product, and as such unclear how often and how much of a quantity would need to be ingested to be clinically impactful. The authors suggest “could be dietary supplement” so should comment on this limitation since minimal data provided in this regard.

Response:

As detailed in the answer to comment eight by the first reviewer, the amount of BL3.2 in the small intestine was not investigated due to limitations of the animal model and the difficulty when translating animal model findings to a relevant human dose. A discussion around previous dosing of V_HHs against rotavirus and ETEC has been added to line 258 and onwards.

8. Authors rightly point out breast milk is protective, but babies suckle many times during the day, and probably require far less antibody since they are not ingesting other things contaminated with V. Cholerae. These limitations should be discussed.

Response:

In a previous study (referred to in the answer to comment seven above), three-daily doses of an anti-rotavirus V_HH was sufficient to significantly reduce stool output of infants infected with rotavirus. This has been included on line 257.

9. 68% blocking, and 27 fold decrease cAMP hard to put into context.

Response:

While it is difficult to directly compare the toxin-neutralizing ability of various antibody formats, we have added a reference to previous work by Das et al. and

their ability to reduce intracellular cAMP 3-fold on line 237 to better put these numbers into context⁹.

10. Appears that 25-50 mcg of CT administered to mice per my calculations, that is a large load that often kills neonatal mice within 18-48 hours. Why did no mice die in the BSA control group? If desire was to harvest/sacrifice before death then would state, but surprising no death at 22 hours. Were control mice moribund at that stage?

Response:

Mice in the BSA control group were moribund at 22 hours and the experiment had to be terminated for ethical reasons. This has been addressed in a previous comment and information has been added to line 575.

11. Why was BSA used as the control and not a non-specific bivalent camelid V_HH in the animal studies. Would mention as limitation/weakness.

Response:

A discussion around additional controls for the in the in vivo studies has been included in the response to the first comment by the first reviewer. In short, considering the stark contrast between BL3.2, the non-specific V_HH, and even the anti-CTXB V_HH now presented in for example Fig. 1b and Supplementary Fig. 4, we argue that the protective effect expected from a non-specific V_HH is comparable to administration of a generic protein such as BSA.

12. The animal studies are supportive but cohorts of animals are small (5 or 6).

Response:

As we did not know how big the differences between the groups would be, we started out with relatively small cohorts (5-6 animals each). However, the initial experiments showed such a strong phenotype that we were able to calculate statistical significance with the relatively small numbers and did not need further animals. In line with our ethics guidelines, we did not add more animals to the groups.

13. Only one animal model used; would mention as limitation; for instance could have evaluated direct challenge in rabbit ileal loops.

Response:

The selection rationale behind using the infant mice model, as asked for by the first reviewer in comment number seven, has been emphasized on line 168.

14. Mention should be made that a product like VHH may have its best utility as a prophylactic for individuals at immediate risk of cholera, and could supplement control programs for such populations such as distributing chlorine tablets (also a very cheap intervention for high risk populations). Mention should be made that CT is a very potent enterotoxin and once internalized by an intestinal epithelial cell that the window of any protection afforded by a VHH to that cell has passed; as such, one might expect a limited role in therapy of cholera patients as opposed to preventing disease before it occurs) although no data currently exist (would explicitly state to assist reader).

Response:

The potential utility of BL3.2 for individuals at immediate risk of infection, as a protective strategy, has been emphasized on 264.

Overall, well written study whose central conclusions regarding initial development and evaluation of a product that disrupts CT-GM1 interaction are supported by the data provided.

Response:

Thank you.

Reviewer #3

The authors generate and characterize an orally deliverable bivalent VHH construct (BL3.2) that binds to the B-subunit of cholera toxin (CTX). This VHH construct interferes with interaction between CTX and receptor GM1, which is critical for the gastrointestinal pathogen *V. cholerae*. The authors show the engineered bivalent is stable under physiological gastrointestinal passage via in vitro studies, as well as active in functional mouse studies. Such a reagent is low cost with potential high payoff in terms of potential impact on populations suffering from cholera. The paper should be of interest to those in the field, as well as the general audience. The manuscript is well-written with clear data/figure presentations.

Response:

Thank you for the positive assessment of our study.

- 1. Not clear why authors are not seeing near 100% blocking, based on assessment of high affinity of BL3.1 and BL3.2. What's known about the affinity of the CTX/GM1 interactions? It would be beneficial for a sentence or two be included.**

Response:

In the revision process, we discovered an incorrect determination of protein concentration used when analysing the CTXB-blocking capacity of BL3.1 and BL3.2. Figure 1 has now been updated to display 100% blocking for BL3.2 at a 1:1 molar ratio (V_HH-V_HH:CTXB).

The affinity of CTXB towards GM1 is reported to be in the nanomolar range (43 nM), comparable to the BL3.1/CTXB interaction. This information has been included on line 41, with a reference to previous work by Turnbull et al¹⁰.

- 2. Significant figures, such as Supplementary table 1, should be reported to the highest least significant digit. For instance k_{on} should be 3.69 ± 0.03 (i.e., the part in parenthesis of the error is not relevant 3.11)**

Response:

Supplementary Table 1 has been updated according to the comment made by the reviewer.

- 3. Methods section: provide methods on how VHH were produced for screening after subcloning into pSANG10 vector.(20) It is not apparent how authors performed expression/purification of 380 monovalent VHH constructs. In addition, reference 20 seems to be an intermediate reference and actual reference to pSANG is C.D. Martin, G. Rojas, J.N. Mitchell, K.J. Vincent, J. Wu, J. McCafferty, D.J. Schofield. A simple vector system to improve performance and utilisation of recombinant antibodies ?? Reading results/methods, it sounds like the VHH clones were expressed to the supernatant/culture media, which doesn't seem to be the case. Details should be present in the methods.**

Response:

The 380 monovalent VHH constructs used for screening purposes were analysed directly in the *E. coli* supernatant, and not purified. This approach relies on the inability of the bacteria to retain all V_HH constructs in the intended periplasmic space, often referred to as “leakiness”. Line 329 and onwards in the method section has now been updated, with a reference to a previous publication using the same screening method⁵.

We have added the correct reference to the pSANG vector on 320. Furthermore, a missing reference on line 332 has been added.

4. DELFIA- provide full name before abbreviation

Response: The full name is provided the first time the assay is mentioned on line 210.

5. Methods: thermal stability experiments: conditions not clear e.g., buffer, pH?

Response:

Details of the buffer (PBS, pH 7.4) have been added to line 432 in the method section.

6. Thermal stability experiments suggested the bivalent BL3.2 possesses a significantly higher melting point (more than 15 degree C) than the single VHH. If true, this would suggest some sort of VHH:VHH interaction. However, it is more likely there may be something up with the analysis or data. As the raw data are not included, I cannot comment. Raw data should be included in supplemental, as well as something describing what the authors believe is going on. If true, it would be of interest.

Response:

The thermal stability of the monomeric V_HH (BL3.1) was never tested, the thermal stability of BL3.2 was only compared to that of CTX (Fig. 2b). Melt curve plots for the thermal stability assay have now been included (Supplementary Fig. 3).

7. SPR data analysis of BL3.1: thank you for including data/fits/residuals. Model/Data not the best fit, but consistent with what appears to be significantly high affinity (near low nM). Suggestion for the future....perform such concentration studies with replicates.

Response:

We are sorry for the confusion. Supplementary Table 1 was incorrectly labelled as SPR data, when in fact only BLI data was presented. Now, SPR analysis of BL3.1-CTXB interaction has been performed and Supplementary Table 1 updated to include new data. These results are presented on line 87. Supplementary Fig. 2 has also been updated to include SPR raw data, and the method section has been updated to include SPR analysis (line 446 and onwards).

8. Authors state: “ BL3.2 achieved half-maximal relative inhibitory concentration (IC50) at 1.5 nM, corresponding to a VHH-VHH:CTX molar ratio of 13:1.” Seems like a strange way to compare molar ratios, as VHH-VHH is a dimer and CTX is a pentamer (based on likely binding site). This would work out to a 6.5 to 5 or roughly 1:1 VHH to CTXB ratio.

Response:

We argue that the 13:1 molar ratio is of interest when discussing potential dosing requirements. However, emphasizing the binding site ratio makes it easier to visualize the mechanistic interaction between BL3.2 and CTX. Therefore, the sentence on line 121 has been expanded to also detail the V_HH:CTXB binding site ratio. Furthermore, the Methods section (line 281 and onwards) has been updated to emphasize the different toxin structures: the monomeric B-subunits used for theoretical calculations of molar ratios, the CTXB pentamer observed in solution, and the full CTX, including the catalytic CTXA, used for cell line and in vivo studies.

9. Binding site predictions: authors state “Molecular dynamics simulations of Model 4 defined the epitope-paratope interaction at high resolution.” The “at high resolution” sounds really strange for a computational model, as all models will provide high resolution (x,y,z coordinates). Please remove high resolution.

Response:

The referred phrasing has been removed and the following sentence has been re-written.

References

1. Yadav, V., Varum, F., Bravo, R., Furrer, E. & Basit, A. W. Gastrointestinal stability of therapeutic anti-TNF α IgG1 monoclonal antibodies. *Int. J. Pharm.* **502**, 181–187 (2016).

2. Goldman, E. R. *et al.* Facile Generation of Heat Stable Antiviral and Antitoxin Single Domain Antibodies from a Semi-synthetic Llama Library. *Anal. Chem.* **78**, 8245–8255 (2006).
3. Sit, B., Fakoya, B. & Waldor, M. K. Animal models for dissecting *Vibrio cholerae* intestinal pathogenesis and immunity. *Curr. Opin. Microbiol.* **65**, 1–7 (2022).
4. Maffey, L., Vega, C. G., Miño, S., Garaicoechea, L. & Parreño, V. Anti-VP6 VHH: An Experimental Treatment for Rotavirus A-Associated Disease. *PLOS ONE* **11**, e0162351 (2016).
5. Rodriguez Rodriguez, E. R. *et al.* Fit-for-purpose heterodivalent single-domain antibody for gastrointestinal targeting of toxin B from *Clostridium difficile*. *Protein Sci.* **33**, e5035 (2024).
6. Jenkins, T. P. *et al.* Protecting the piglet gut microbiota against ETEC-mediated post-weaning diarrhoea using specific binding proteins. *Npj Biofilms Microbiomes* **10**, 1–15 (2024).
7. Fiil, B. K. *et al.* Orally active bivalent VHH construct prevents proliferation of F4+ enterotoxigenic *Escherichia coli* in weaned piglets. *iScience* **25**, 104003 (2022).
8. Rivera-Chávez, F. & Mekalanos, J. J. Cholera toxin promotes pathogen acquisition of host-derived nutrients. *Nature* **572**, 244–248 (2019).
9. Das, S. *et al.* Neutralization of cholera toxin with nanoparticle decoys for treatment of cholera. *PLoS Negl. Trop. Dis.* **12**, (2018).
10. Turnbull, W. B., Precious, B. L. & Homans, S. W. Dissecting the Cholera Toxin–Ganglioside GM1 Interaction by Isothermal Titration Calorimetry. *J. Am. Chem. Soc.* **126**, 1047–1054 (2004).

REVIEWERS' COMMENTS

Reviewer #1 (Remarks to the Author):

Thank you for incorporating many of the suggested changes, including significant improvements to the mechanistic understanding via HDX-MS data and generation of mutants. I am still of the opinion a control VHH-VHH and/or the anti-CTX VHH from Goldman et al 2006 (formatted as a VHH-VHH) would have been preferred to BSA in the in vivo animal experiments; however, I accept the arguments put forth in the rebuttal.

Minor comments:

Line 97-98: please change "...with a KD determined to be 0.76 nM.." to "...with an apparent KD determined to be 0.76 nM...". Also change "...and to be 85.5 nM..." to "...and a monovalent KD of 85.5 nM..." to reflect the avid binding event occurring in the first assay orientation. It is also fine to omit the word "monovalent", if the authors prefer, when referring to the second affinity value

Response:

Line 97-98 changed accordingly.

Supp Fig 10: the complex formation by SEC is helpful, although SEC-MALS would have been better suited. Can SEC MW standards be labelled on the chromatogram in an attempt to guide the reader? (I recognize this isn't analytical SEC, but even some reference point with MW stds will help with interpretation of peaks/complexes).

Response:

Supplementary Figure 10 (and methods line 520) has been updated to contain a secondary panel (Fig. 10b), where an overlay of separate SEC runs of BL3.2 and BL3.1 has been added. With this new SEC data for BL3.2 and BL3.1, it is hopefully clearer that the observed CTXB aggregates caused by BL3.2 is of a much larger molecular size than the two individual V_HH constructs.

The reason for not using a reference point with MW stds is due to the shape of the pentameric CTXB. In cases where the proteins are perfectly globular, the estimation of MWs from calibration curves on SEC could give a reasonable approximation of the size. However, due to the "flattened sphere" shape of CTXB, this approach gives us false protein size estimates.

Editorial note: this reviewer was also asked to comment in place of reviewer 2 who was unable to provide a response at this time:

I have reviewed the responses to R2 and believe all concerns have been adequately addressed.

Minor comments to Authors:

1. Sentence on lines 241-243 needs to be edited. As written, it suggests single-domain antibodies can be derived from IgAs

Response:

The sentence on line 241-243 has been re-written.

2. Please explicitly state limitations of this study in the Discussion in 1-2 sentences
 - small animal cohort for in vivo testing
 - use of BSA instead of control VHH (irrelevant or anti-CTX from Goldman) in the in vivo expts
 - dosing not examined/optimized due to limitations of model

Response:

The small animal cohort used in this study and the lack of an additional control have been emphasized on line 230 in the discussion.

The limitations of the model with regards to identifying the human equivalent dose has been emphasized on line 262 and onwards.

Reviewer #3 (Remarks to the Author):

The modifications addressing all reviewer comments have improved the manuscript. I am glad to see that revisiting the binding/blocking experiments identified conc. issues that when corrected, now provide a more consistent set of data. Please see a few additional comments/suggestions.

1. Methods

The authors should be more explicit in their methodology. Errors were their protein concentrations were uncovered during their revisions. How specifically were VHH concentrations determined? The authors state via 280 nm absorbance. Please state the 280 nm extinction coefficients. I'm assuming values from something like (Protparam: <https://web.expasy.org/protparam/> were used). This would be the most accurate and accessible method. It also appears a BCA assay was used to determine concentration for affinity measurements. Any particular reason why the BCA assay is being used (which is more appropriate for total heterogeneous protein concentration) rather than direct (more accurate) 280 extinction coefficient for this quantitative measurements? I would not be surprised if differences in conc. were stemming from BCA from one time/individual to another.

Response:

The extinction coefficients for BL3.1 and BL3.2 are now included in the Mendeley Data (linked in the manuscript). The BCA assay was used as a complementary protein determination assay when V_HH construct were produced using the K. phaffii expression system, to reduce the potential influence on other components in the yeast supernatant on 280 nm absorbance.

2. Results: line 89 "BL3.1 displayed high affinity for CTXB, with a KD determined to be 0.76 nM by BLI (with BL3.1 as ligand) and to be 85.50 nM by SPR (with CTXB as ligand) (Supplementary Table 1). These values are similar to the KD (77 nM) of a previously reported anti-CTX VHH 30."

- This is a 100-fold difference. Since this is almost certainly due to avidity effects with the way the authors setup the initial BLI experiments, they should be direct and state this well-known issue for the likely difference in K values.

Response:

As mentioned in the response to reviewer #1, line 89 has been re-written to be specific about the apparent and monovalent KD without speculating about the differences in KD values.

3. Thermal denaturation experiments: Thank you for the clarification and including the raw data in the Suppl. Info. Based on the authors assessment: “Protein thermal stability screening using differential scanning fluorimetry (Protein Thermal Shift™) showed that BL3.2 had even greater thermal stability (67.3 °C) than CTX (52.0 °C) (Fig. 2b and Supplementary Fig. 3)” Examining the raw differential scanning fluorimetry data, the T_m of 67.3 for BL3.2 looks accurate. This is not the case for the CTX runs. There is just not enough of a change in signal between native to say the T_m is 52. Almost appears if T_m was identified without looking at data, but derivative signal alone. If anything, based on the general expected trend for DSF data, there is a small increase in signal right before the max raw signal of ~72dC, which would suggest the mid-point is very close to ~67dC, suggesting almost identical melting temperatures between the two VHH constructs. At a minimum, I’d suggest using the max derivative peaks which occur right around 67dC. Even better, signal response can be optimized by varying protein concentration and dye to get the best signal change to increase confidence in the observed T_m value.

Response:

Thermal stability data presented on line 107 has been updated to correspond to the derivative T_m . With this change, there is a smaller difference in thermal stability between BL3.2 (71.5 °C) and CTX (67.9 °C). Line 438 in Methods has been updated to specify that the derivative T_m has been calculated.

4. Suppl Figure 10: appears to be missing the free BL3.1 and BL3.2 species profile to aid interpretation.

Response:

Free BL3.1 and BL3.2 species profile have been added to Supplementary Figure 10b.